# Empowering Multimodal Understanding Model with Interleaved Multimodal Generation Capability

**Jianwen Sun[1,2,*]**  **Yukang Feng[1,2,*]**  **Chuanhao Li[3]**  **Fanrui Zhang[2]**  **Zizhen Li[1,2]**

**Jiaxin Ai[2]**  **Sizhuo Zhou[2]**  **Yu Dai[1,†]**  **Shenglin Zhang[1]**  **Kaipeng Zhang[2,3,†]**

[1]Academy for Advanced Interdisciplinary Studies, Nankai University, Tianjin, 300071, China
[2]Shanghai Innovation Institute (SII), Shanghai, China    [3]Alaya Lab

[*]Equal contribution.    [†]Corresponding authors.
`sunjianwen@mail.nankai.edu.cn, daiyu@nankai.edu.cn, kaipeng.zhang@shanda.com`

Reviewed on OpenReview: `https://openreview.net/forum?id=4TLXaJt8Rq`

## Abstract

Unified multimodal understanding and generation have attracted much attention in the field of vision and language in recent years. Existing unified models (UniMs) aim to simultaneously learn understanding and generation capabilities, which require a large amount of computational resources and have defects in two aspects: 1) difficulty in generating interleaved text-image content; 2) weaker understanding capabilities than multimodal large language models (MLLMs). To bridge this gap, we propose ARMOR, a resource-efficient framework designed to "upgrade" rather than "retrain from scratch" expert MLLMs. Our core principle is to endow MLLMs with generation capabilities while preventing catastrophic forgetting of their top-tier understanding capabilities. We achieve this goal through three key innovations: (1) an asymmetric architecture that isolates a lightweight generative decoder from the frozen MLLM core via a forward-switching mechanism to enable seamless interleaved generation; (2) a meticulously curated high-quality interleaved dataset; (3) a progressive "What or How to Generate" (WoHG) three-stage training algorithm. Experiments demonstrate that ARMOR successfully upgrades a leading MLLM, retaining over 95% of its original understanding performance while achieving reasonable image generation quality at less than 1/70 the cost of training from scratch. This demonstrates the effectiveness of our core idea: "the efficient paradigm of upgrading and expanding existing expert MLLMs into UniMs."

## 1  Introduction

Unified understanding and generation is a key direction in the development of vision-language models, requiring models to handle both understanding and generation tasks. Existing Unified Models (UniMs), such as Show-oXie et al. (2024), Emu3Wang et al. (2024b); Sun et al. (2023) and Janus-proChen et al. (2025); Wu et al. (2024a), are designed to learn multimodal understanding and generation capabilities simultaneously. However, their training demands substantial computational resources, which significantly hinders their scalability and the ability to make personalized modifications. Furthermore, most of these models adopt two independent output modes for unified capability integration, making it difficult to generate continuous, interleaved text-image content. Therefore, how to efficiently enable interleaved text-image output is one of the key problems we address.

Another critical issue is that the multimodal understanding capabilities of these from-scratch UniMs exhibit a significant performance gap compared to Multimodal Large Language Models (MLLMs) specialized for understanding tasks, such as Qwen-VLBai et al. (2023); Wang et al. (2024a); Bai et al. (2025) and

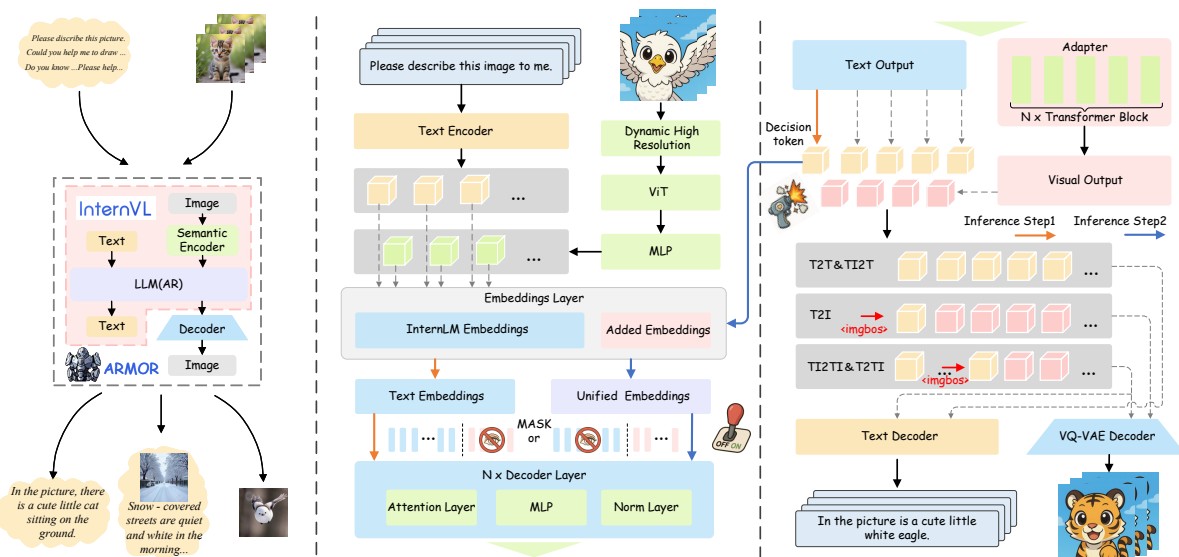

Figure 1: Schematic Diagram and Workflow of the ARMOR Framework: Input unimodal or mixed-modal content, and the first text token (decision token) determines the subsequent output modality. Special tokens are used to distinguish decoder bases to achieve output of content in different modalities.

InternVL-2.5Chen et al. (2024). This observation raises a core research question: can we efficiently extend the generative capabilities of a powerful, pre-existing understanding model, rather than starting from scratch? The key technical challenge of this approach lies in how to introduce new capabilities while avoiding significant degradation to the model's original, powerful understanding abilities-a problem widely known as "catastrophic forgetting."

To address the aforementioned challenges, we propose the ARMOR framework. It serves as a highly modular and non-invasive solution designed to provide a capability upgrade framework, demonstrated on InternVL2.5, whose additive design is architecturally compatible with mainstream MLLMs sharing similar encoder-connector-LLM structures. Architecturally, we connect a lightweight image decoder to the original MLLM and establish a corresponding forward-switching mechanism, which protects the model's original capabilities at an architectural level. Regarding training, we propose a three-stage training method that guides the model to master high-quality image generation through progressive learning, while effectively preventing catastrophic forgetting.

Our main contributions are as follows: (1) We propose a resource-efficient "upgrade" paradigm for building unified models, offering a new path distinct from training from scratch. (2) We design a complete technical solution that includes an asymmetric architecture, a forward-switching mechanism, and the WoHG training algorithm to address the problem of learning new capabilities while preserving the original ones. (3) We validate the effectiveness of the proposed framework through extensive experiments and demonstrate its potential in achieving interleaved text-image generation.

## 2 Related Work

### 2.1 Multimodal Understanding

CLIPLi et al. (2023a); Radford et al. (2021) pioneered cross-modal alignment via contrastive learning, greatly inspiring the subsequent development of MLLMs. Currently, two dominant strategies have emerged for aligning vision encoders with large language models. The first is explicit attention interaction, as seen in BLIP-2's Q-FormerLi et al. (2023a) and Flamingo's cross-attentionAlayrac et al. (2022), which project visual features into text-aligned soft tokens through learnable queries to achieve deep vision-language fusion. The

second is implicit space mapping, exemplified by InternVLBai et al. (2025); Chen et al. (2024) and Qwen-VLWang et al. (2024a); Bai et al. (2023), which utilize lightweight MLP networks to directly map visual features into the input embeddings of the language model. Although these MLLMs differ in their specific designs, they share core components and continue to evolve through improved visual representations and more efficient alignment methods. The primary objective of these models is to achieve deep multimodal understanding, but their architectures inherently lack the capability to generate visual content such as images.

## 2.2 Visual Generation

The field of visual generation has evolved through autoregressive and diffusion-based approaches. Early autoregressive modelsParmar et al. (2018); Esser et al. (2021a); Ravuri & Vinyals (2019); Chen et al. (2020) utilized Transformers for pixel-level text-to-image synthesis but faced challenges with high-resolution fidelity. Subsequent token-based methodsSun et al. (2024) improved sequence prediction but remained limited in detail. In contrast, diffusion models such as the Stable Diffusion seriesRombach et al. (2022a); Esser et al. (2024); Podell et al. (2023), DALL-E 2Ramesh et al. (2021; 2022); Betker et al. (2023), and FLUXLabs (2024) have gained prominence by iteratively denoising images to achieve state-of-the-art visual quality. These methods excel at generating high-fidelity, detailed outputs for text-to-image tasks; however, their frameworks are primarily designed for generation and lack the capacity for deep understanding and complex reasoning about image content.

## 2.3 Unified Understanding and Generation

In recent years, building models that can unify understanding and generation has attracted significant research interest. Existing approaches can be broadly categorized. Some works, like Next-GPTWu et al. (2024b); Zhou et al. (2024a), achieve unified functionality by connecting separate understanding and generation systems, but they are not end-to-end native unified models. Other models, such as ChameleonTeam (2024) and Emu2Wu et al. (2024d); Tian et al. (2024); Esser et al. (2021b); Zhuang et al. (2025); Van Den Oord et al. (2017), adopt end-to-end autoregressive architectures that tokenize images and predict them within the same sequence as text. Despite the significant progress these models have made in unified multimodal capabilities, they generally face two major challenges. First, the joint training from scratch requires substantial computational resources. Second, and more critically, the understanding capabilities of these unified models often cannot match those of specialized MLLMs, exhibiting a clear performance gap. These challenges motivate our exploration of a new paradigm for building such models, one that preserves the original capabilities of expert models. Recent works such as MetaMorph Tong et al. (2024) and Orthus Kou et al. (2025) also explore unified models, but the former updates the LLM backbone during training, while the latter is trained from scratch. ARMOR instead keeps the entire MLLM frozen, offering a more non-invasive upgrade path.

## 2.4 Model Capability Expansion

Extending a model with new capabilities without compromising its existing ones is a long-standing challenge in machine learning, a problem known in the field of Continual Learning as "catastrophic forgetting"McCloskey & Cohen (1989). Parameter-Efficient Fine-Tuning (PEFT) techniques offer an effective approach to address this issueParisi et al. (2019); Wu et al. (2024c); Hu et al. (2021). PEFT methods like LoRA and Adapters freeze the backbone parameters of a pre-trained model and train only a small number of new, pluggable modules to adapt to new tasksHoulsby et al. (2019a;b). This approach not only significantly reduces training costs but also effectively mitigates catastrophic forgetting due to its non-invasive nature. However, existing PEFT methods primarily focus on adapting models to new downstream tasks or domains (Task/Domain Adaptation)Zhou et al. (2024b); Ye et al. (2025a). In contrast, our ARMOR framework applies this core "freeze-and-extend" idea to a more complex and different scenario: adding an entirely new output modality (i.e., image generation) to a MLLM, while keeping the original backbone entirely frozen to maximally preserve its understanding capabilities.

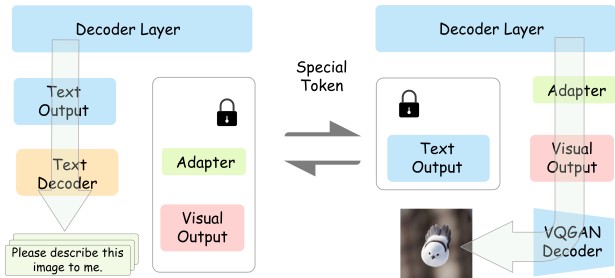

Figure 2: Schematic Diagram and Workflow of the ARMOR Framework

## 3 Framework

### 3.1 Architecture

To validate our core thesis-that it is feasible to efficiently expand a pre-trained MLLM into a unified model-we introduce the ARMOR framework. This framework follows the core philosophy of "upgrading" rather than "rebuilding," aiming to empower powerful, existing MLLMs with high-quality image generation capabilities in a non-invasive and lightweight manner, thereby providing a complete technical implementation to demonstrate the feasibility of our thesis.

As depicted in Figure 1, the framework operates by attaching a set of trainable, modular components to a frozen MLLM backbone through additive operations. All modifications to the MLLM are strictly additive, a key prerequisite for ensuring feasibility, as this guarantees minimal impact on the original model. These components primarily consist of an expanded vocabulary and its corresponding embedding layer, a lightweight adapter, and a pre-trained image decoder(from the Chameleon model). The content of the expanded vocabulary is shown in Table 1. We use the ARMOR framework to upgrade the InternVL2.5-8B model, and this model will be referred to as ARMOR hereafter. Regarding the parameter scale, all newly introduced trainable components total approximately 0.9B parameters, which is only about 10% of the 8B backbone model. Among them, the expanded embedding and output layers contribute 0.035 B each (total 0.07 B), while the Transformer adapter accounts for the remaining 0.83 B. This lightweight design is intended to demonstrate the practical feasibility of this technical path in terms of both cost and efficiency.

To fully demonstrate the feasibility of this approach, the following subsections will provide a detailed analysis of the two key challenges in implementing the ARMOR framework: how we address the technical challenge of interleaved text-image generation (Section 3.2), and how we overcome the core challenge of preserving original capabilities while introducing new ones (Section 3.3).

| Token Type | Head | Functional Description |
|---|---|---|
| *Special Tokens:* | | |
| `<imgbos>` | text output | Switch model to visual output mode, begin image generation |
| `<imgend>` | visual output | Terminate image generation, revert to text output mode |
| `<imgpad>` | visual output | Padding placeholder in image token sequences |
| *Image Content Tokens:* | | |
| 8192 image tokens | visual output | Content representation tokens |

Table 1: Special token and image content token specifications.

### 3.2 Efficient Interleaved Generation

To address the technical challenges of interleaved text-image generation and optimize output efficiency within a unified autoregressive framework, we propose two key architectural designs: an asymmetric encoder-decoder structure and an explicit gated switching mechanism.

The "asymmetry" in our design lies in our full reuse of the MLLM's original, well-trained vision encoder for image understanding, while attaching a new, lightweight image decoder exclusively for the generation task. The fundamental hypothesis behind this decoupled architecture is that a powerful pre-trained encoder is sufficient to acquire competitive image generation at low cost, which is the goal of this work, rather than to maximize absolute generation fidelity, thereby obviating the need to design or train a symmetric, tightly-coupled encoder specifically for the generation task. This design serves as our guiding principle for implementing a non-invasive upgrade.

This mechanism is key to achieving seamless interleaved generation and represents a core distinction from existing models. Unlike other models that might use soft weights or perform joint prediction over a shared vocabulary, our mechanism is a hard, exclusive gating system, the workflow of which is illustrated in Figure 2. We establish separate output heads for the text and image modalities. During the autoregressive generation process, the model uses two special control tokens, `<imgbos>` and `<imgend>`, to act as explicit "switches" for modality transition.

The objective of introducing this mechanism is to address two potential challenges during the output process. First, we aim to avoid inefficient Softmax computation over a joint vocabulary that includes text tokens and thousands of image tokens. Second, by physically separating the two output heads, we aim to architecturally eliminate the "long-tail competition" problem between text and image tokens, with the expectation of achieving more stable and efficient training dynamics. Formally, this decouples the joint probability over a shared vocabulary into two independent distributions, $P(\text{text}|x)$ and $P(\text{image}|x)$, preventing gradient interference between modalities. The effectiveness of these design hypotheses will be validated in the experiments section through analysis of convergence speed and final performance.

Below, we use Algorithm 1 to precisely describe this generation process, clearly illustrating how the model dynamically switches between text and image modalities.

---

**Algorithm 1:** Interleaved Sequence Generation Algorithm of ARMOR

---

**Input:** Multimodal context $X$, maximum generation length $L_{\max}$
**Output:** Generated sequence $Y$
**Data:** $Y \leftarrow [\texttt{<bos>}]$, current mode $m \leftarrow$ text

**1** **while** $|Y| < L_{\max}$ **and** $Y[-1] \neq \texttt{<eos>}$ **do**
**2**     Obtain LLM's hidden state output $h_t$ based on input $X$ and generated sequence $Y$;
**3**     **if** $m = text$ **then**
**4**        Compute probability distribution $P_t \leftarrow \text{Softmax}(\text{Head}_{\text{text}}(h_t))$;
**5**     **else**
**6**        Compute probability distribution $P_t \leftarrow \text{Softmax}(\text{Head}_{\text{image}}(h_t))$;
**7**     Sample the next token $y_{t+1} \leftarrow \text{Sample}(P_t)$;
**8**     Append $y_{t+1}$ to sequence $Y$;
**9**     **if** $y_{t+1} = \texttt{<imgbos>}$ **then**
**10**        $m \leftarrow$ image ;                     // Switch to image generation mode
**11**     **else if** $y_{t+1} = \texttt{<imgend>}$ **then**
**12**        $m \leftarrow$ text ;                       // Switch back to text generation mode
**13** **return** *Complete sequence $Y$*

---

### 3.3 Capability Expansion

A core challenge is how to enable the model to learn new capabilities while effectively avoiding catastrophic forgetting. Inspired by the field of Continual Learning, our solution avoids end-to-end joint training and instead employs a three-stage curricular training algorithm named "What-or-How-to-Generate" (WoHG). The core of this algorithm lies in protecting existing knowledge by decomposing the learning objective and isolating model parameters.

**Learning Objectives**

To enable flexible control over different learning tasks, we introduce a weighted loss calculation method that allows for dynamic adjustments across the training stages. We formalize the training goal into two independent learning objectives: text prediction and image prediction.

*(1)Text Prediction Loss* The loss for text generation, $\mathcal{L}_{\text{text}}$, is defined as the standard autoregressive cross-entropy loss, computed only over the text token prediction steps:

$$\mathcal{L}_{\text{text}} = -\sum_{t=1}^{T} \mathbb{I}_{\text{text}}(t) \cdot \log P_{\text{text}}(y_t \mid y_{<t}, X) \tag{1}$$

Here, $T$ is the total length of the target sequence. The indicator function $\mathbb{I}_{\text{text}}(t)$ is 1 if the target token $y_t$ at timestep $t$ is a text token, and 0 otherwise. $y_{<t}$ represents the sequence of tokens preceding timestep $t$, $X$ is the multimodal input, and $P_{\text{text}}$ is the probability distribution from the text output head.

*(2)Image Prediction Loss* Similarly, the loss for image generation, $\mathcal{L}_{\text{img}}$, is defined as the cross-entropy loss computed over the image token prediction steps:

$$\mathcal{L}_{\text{img}} = -\sum_{t=1}^{T} \mathbb{I}_{\text{img}}(t) \cdot \log P_{\text{image}}(y_t \mid y_{<t}, X) \tag{2}$$

where the indicator function $\mathbb{I}_{\text{img}}(t)$ is 1 only when the target token $y_t$ is an image token. $P_{\text{image}}$ is the probability distribution from the image output head.

*(3)Overall Learning Objective* The final objective is a weighted sum of the text and image losses. The weights, $\alpha$ and $\beta$, serve as curriculum control parameters that are dynamically adjusted during the WoHG training stages to shift the model's learning focus.

$$\mathcal{L}_{\text{total}} = \alpha \mathcal{L}_{\text{text}} + \beta \mathcal{L}_{\text{img}} \tag{3}$$

**Training Datasets** ARMOR must learn *when* to answer with text, *when* to answer with images, and *when* to interleave both. Accordingly, we organise the training corpus into four dedicated subsets.

*(1)Text–to–Text (t2t).* Purpose: teach pure language dialogue and VQA style answers that do *not* involve image generation. Source: 100 k self-curated instruction pairs covering daily conversation, reasoning, and factual Q&A.

*(2)Text+Image–to–Text (ti2t).* Purpose: strengthen multimodal *understanding*; the model sees an image but should reply only with text. Source: 300 k high-quality samples selected from ShareGPT-4V, LLaVA, and distillations of InternVL-2.5.

*(3)Text–to–Image (t2i).* Purpose: explicit image generation given a prompt. Source: 2.5 M captions obtained from (a) our own synthetic prompts, (b) LAION-nolang-aesthetics-27M filtered by CLIP score and aesthetics, (c) the Text-to-Image-2M collection.

*(4)Text–to–Text+Image (t2ti).* Purpose: train the model to produce interleaved text–image replies that mimic human communication. Source: 2.5 M pieces of data expanded based on InterSynFeng et al. (2025); Li et al. (2024b) where each answer contains both a short textual explanation and one illustrative image.

The four subsets are used at different stages of the WoHG curriculum, enabling the model to master modality selection, image rendering, and mixed-output refinement in a progressive manner. All datasets will be released to the community upon publication.

**WoHG Training Algorithm** We employ a three-stage curriculum called **W**hat-or-**H**ow-to-**G**enerate (WoHG). Each stage targets a distinct skill and unfreezes only the parameters required for that skill, thus keeping training economical and avoiding catastrophic forgetting (see Figure3).

*Stage 1 "What to generate?"* Objective: Teach the model to choose the proper response modality: text, image, or a mixture. Data: t2t 100 k, ti2t 300 k, t2i 100 k, t2ti 100 k. Loss setting: text weight $\alpha = 1$, image weight $\beta = 0$. Trainable parts: special-token embeddings and the modality switch (Figure 3a).

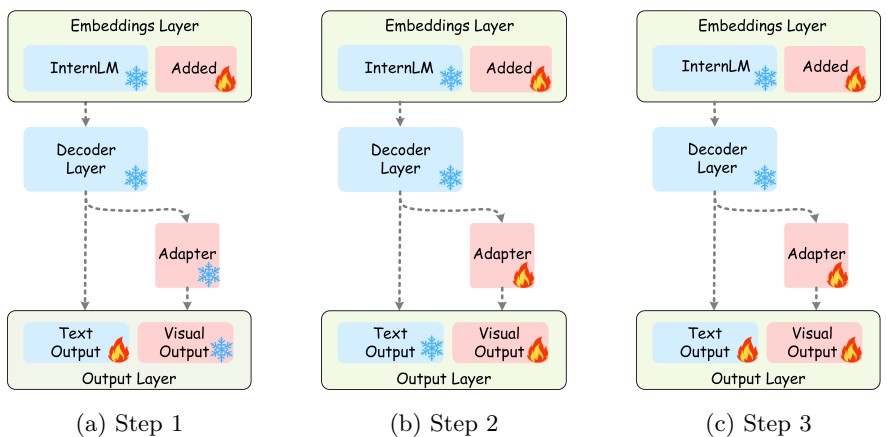

Figure 3: WoHG training algorithm.

*Stage 2 "How to generate?"* Objective: Activate the image branch and align VQ codes with the linguistic context. Data: t2i 2.5 M and t2ti 2.5 M. Loss setting: $\alpha = 0$, $\beta = 1$. Trainable parts: adapter layers, image-token embeddings, and the image output head (Figure 3b).

*Stage 3 "How to answer better?"* Objective: Refine text–image coherence and recover any minor loss in understanding accuracy. Data: t2t 50 k, ti2t 300 k, t2i 300 k, t2ti 50 k. Loss setting: $\alpha = 1$, $\beta = 1$. Trainable parts: same as Stage 2 with the text output head unfrozen (Figure 3c).

*Optimisation details.* All stages use AdamW with $\beta_1 = 0.9$, $\beta_2 = 0.999$, cosine learning-rate decay, gradient clipping at 1.0, and weight decay 0.05. Peak learning rates are $4 \times 10^{-5}$ in Stage 1, $1 \times 10^{-4}$ in Stage 2, and $5 \times 10^{-5}$ in Stage 3. Training is conducted on eight H100 GPUs. Further implementation details and per-stage data ratios are reported in the supplementary material.

Upon completing WoHG, ARMOR retains the original MLLM's comprehension capability while gaining the power to emit truly interleaved text–and–image sequences within a single autoregressive stream. A detailed analysis of alternative training configurations and stage boundaries is provided in Appendix H.

## 4 Experiments

### 4.1 Settings

**(1) Evaluation Dimensions**

We assess ARMOR along three key axes: multimodal understanding, visual generation, and interleaved generation capability.

**(2) Benchmarks**

**Multimodal Understanding.** We follow the VLMEvalKit protocol and report results on nine standard datasets: MMMUYue et al. (2024), MME-PFu et al. (2023), MMEFu et al. (2023), MMVetYu et al. (2023), MM-BenchLiu et al. (2025), SEEDBench-imgLi et al. (2024a), HallusionBenchGuan et al. (2024), POPELi et al. (2023b), and LLaVABenchLiu et al. (2023b). Together, these datasets cover tasks such as knowledge reasoning, fine-grained recognition, hallucination detection, and open-ended dialogue.

**Visual Generation.** For text-to-image generation, we use GenEval, a unified suite for automatic and human evaluation. We further compute the FIDHeusel et al. (2018) score on 30,000 prompts randomly selected from the MS-COCOLin et al. (2015) dataset to measure distributional fidelity.

**Interleaved Generation.** To evaluate the capability of interleaved text-image generation, we introduce two metrics: (1) Switch-Accuracy: Measures the accuracy of the model in selecting the correct output modality. We test on a set of 900 prompts, with 300 prompts corresponding to each of the three ground-truth modalities. The text-only prompts are sampled from ShareGPT, the image-only prompts from GenEval, and the mixed

prompts from InterSyn Feng et al. (2025). The ground-truth modality is determined by the inherent nature of each source dataset, requiring no subjective annotation.

(2) Interleave-CLIPScore: Evaluates the semantic coherence of interleaved outputs. This is tested on a subset of 300 prompts that require interleaved responses, and we report the average CLIP score.

**(3)Baselines** To put ARMOR in context we compare it with all public systems that appear in the comprehensive benchmark table. We categorize these competing models into four natural groups based on their functions and evaluate each model using its official checkpoints and release settings.

First, the **understanding-only** group focuses on perception and reasoning but cannot synthesise images. It contains Qwen-VLBai et al. (2023), Qwen2-VLWang et al. (2024a) and Qwen2.5-VLBai et al. (2025), InternVL 2.5Chen et al. (2024), DeepSeek-VL2Wu et al. (2024e), LLaVA-v1.5Liu et al. (2023b), LLaVA-ov and LLaVA-Next-VicunaLiu et al. (2023a), InstructBLIPDai et al. (2023), Llama-3-VILA 1.5 and Emu3_ChatWang et al. (2024b).

The second group comprises **unified models without interleaved text-image output**. This set includes Show-o-256Xie et al. (2024), SEED-XGe et al. (2024), VILA-U-384Wu et al. (2024d), LWMLiu et al. (2024a), the TokenFlow family (B, L, XL-Vicuna and XL-Qwen), SynerGen-VLLi et al. (2024b), both sizes of Janus-ProWu et al. (2024a); Chen et al. (2025) and MetaMorph Tong et al. (2024).

The third group contains **unified models with interleaved text-image output**. Here we report results for ChameleonTeam (2024), VARGPTZhuang et al. (2025), Orthus Kou et al. (2025) and ARMOR.

Finally, when analysing pure image quality we also reference dedicated **generation-only** systems such as DALL-E 2 / 3, Stable Diffusion v1.5, D-DiTHuang et al. (2025), SD-XL and SD-3, LDMRombach et al. (2022b), LlamaGen, PixArtChen et al. (2023) and Emu3-Gen.

## 4.2 Results

**(1)Quantitative Evaluation** Tables 2, 3, and 5 summarize the quantitative evaluation results of ARMOR on multiple benchmarks. This data strongly supports the effectiveness of our method, primarily demonstrated in the following three aspects: (1)**Effective Prevention of Catastrophic Forgetting.** Compared to its base model, InternVL-2.5, ARMOR retains approximately 95% of the original understanding accuracy after learning the generation capability (e.g., the score on MMMU only slightly drops from 53.5 to 51.5). Furthermore, it achieves significantly higher understanding scores compared to other unified models. This demonstrates the success of our asymmetric architecture and the WoHG curricular training in preserving the model's original core capabilities. (2)**Low-Resource Image Generation.** In terms of image generation quality, ARMOR achieves a GenEval score of 0.51 and an FID of 9.07. While a performance gap remains compared to top-tier specialized generation models, it has reached a comparable level. Achieving this level of performance requires only about 500 A100-GPU days of marginal training cost for adding generation capability, far lower than that of Chameleon (35,687) and Janus-Pro (3,584), showcasing an extremely high cost-performance ratio. (3)**Accurate Interleaved Text-Image Generation.** The results validate the effectiveness of our architecture and data in solving the problem of interleaved generation. In terms of modality selection accuracy, ARMOR can accurately determine the user's intent, performing far better than baselines like VARGPT, which almost completely fails in image generation scenarios. Regarding content coherence, ARMOR also achieves the highest Interleave-CLIPScore, demonstrating the strong semantic alignment of its generated text and images. We further evaluate ARMOR on InterleavedBench Liu et al. (2024b), an established benchmark using GPT-4o for multi-dimensional assessment. As shown in Table 4, ARMOR achieves an average score of 3.34, significantly outperforming end-to-end models such as MiniGPT-5 (1.82) and EMU-2 (1.68), with Text Quality and Text-Image Coherence scores approaching pipeline-based systems. Note that the current version generates at most one image per response; the architecture natively supports multiple image segments, and we leave multi-image generation for future work. (4)**Compositional and Dense-Prompt Generation.** To test whether ARMOR handles longer, compositional instructions rather than only short, simple prompts, we further evaluate it on two challenging benchmarks: GenEval++ Ye et al. (2025b), which applies a GPT-4.1 checklist over object, count, color, position, and size, and DPG-Bench Hu et al. (2024), which uses long, dense multi-object prompts. As shown in Table 6, ARMOR scores

Table 2: Evaluation on multimodal understanding benchmarks. (SEED, Hall, and LLaVA represent SEEDBench-img, HallusionBench, and HallusionBench LLaVABench respectively).

| Method | Par. | MMMU | MME-P | MME | MMVet | MMB | SEED | Hall | POPE | LLaVA |
|---|---|---|---|---|---|---|---|---|---|---|
| *Understanding models* | | | | | | | | | | |
| Qwen2.5-VL | 7B | 56.2 | 1685.2 | 2299.2 | 66.6 | 83.5 | 71.0 | 56.3 | 86.1 | 80.6 |
| InternVL 2.5 | 8B | 53.5 | 1688.2 | 2338.9 | 59.6 | 82.0 | 77.0 | 49.0 | 88.9 | 80.3 |
| Qwen2-VL | 7B | 53.7 | 1639.2 | 2276.3 | 61.8 | 82.8 | 76.0 | 50.4 | 88.4 | 70.1 |
| LLaVA-Next-Vicuna | 13B | 37.3 | 1448.4 | 1745.6 | 44.9 | 70.0 | 71.4 | 31.8 | 87.8 | 73.9 |
| LLaVA-ov | 7B | 47.9 | 1577.8 | 1993.6 | 51.9 | 83.2 | 76.7 | 31.6 | 88.4 | 81.0 |
| Llama-3-VILA1.5 | 8B | 37.4 | 1438.8 | 1698.5 | 41.9 | 62.1 | 65.0 | 35.3 | 83.3 | 71.7 |
| DeepSeek-VL2 | 16B | 54.0 | 1632.7 | 2230.2 | 60.0 | 84.1 | 77.0 | 45.3 | – | 89.7 |
| LLaVA-v1.5 | 7B | 35.7 | 1506.2 | 1808.4 | 32.9 | 66.5 | 65.8 | 27.6 | 86.1 | 61.8 |
| InstructBLIP | 7B | 30.6 | 1137.1 | 1391.4 | 33.1 | 33.9 | 44.5 | 31.2 | 86.1 | 59.8 |
| Qwen-VL-Chat | 7B | 37.0 | 1467.8 | 1860.0 | 47.3 | 61.8 | 64.8 | 36.8 | 74.9 | 67.7 |
| Emu3_Chat | 8B | 33.9 | 1334.1 | 1610.5 | 29.1 | 63.8 | 69.2 | 31.7 | 83.3 | 49.2 |
| *Uni-modal models* **without** *interleaved text-image output* | | | | | | | | | | |
| Show-o-256 | 1.3B | 25.1 | 948.4 | – | – | – | – | – | 73.8 | – |
| SEED-X | 17B | 35.6 | 1435.7 | – | – | – | – | – | 84.2 | – |
| VILA-U-384 | 7B | – | 1401.8 | – | 33.5 | – | 59.0 | – | 85.8 | – |
| LWM | 7B | – | – | – | 9.6 | – | – | – | 75.2 | – |
| TokenFlow-B | 13B | 34.2 | 1353.6 | 1660.4 | 22.4 | – | 60.4 | – | 84.0 | – |
| TokenFlow-L | 13B | 34.4 | 1365.4 | 1622.9 | 27.7 | – | 62.6 | – | 85.0 | – |
| TokenFlow-XL-Vicuna | 13B | 38.7 | 1545.9 | 1840.9 | 40.7 | – | 68.7 | – | 86.8 | – |
| TokenFlow-XL-Qwen | 14B | 43.2 | 1551.1 | 1922.2 | 48.2 | – | 72.6 | – | 87.8 | – |
| SynerGen-VL | 2.4B | 34.2 | 1381.0 | 1837.0 | 34.5 | 53.7 | 62.0 | – | 85.3 | – |
| Janus-Pro | 7B | 41.6 | 1516.7 | 1791.7 | 45.1 | 62.6 | 70.1 | 39.5 | 78.9 | 74.4 |
| MetaMorph | 8B | 41.8 | – | – | – | 75.2 | 71.8 | – | – | – |
| *Uni-modal models* **with** *interleaved text-image output* | | | | | | | | | | |
| Chameleon | 7B | 22.4 | 153.1 | 202.7 | 8.3 | 15.4 | 30.5 | 17.1 | 19.4 | 26.6 |
| VARGPT | 9B | 36.4 | 1488.8 | – | – | 67.6 | 67.9 | – | 84.4 | – |
| Orthus | 7B | 28.2 | 1265.8 | – | – | – | – | – | 79.6 | – |
| **ARMOR (InternVL2.5)** | 8B | **51.5** | **1635.2** | **2281.5** | **56.3** | **78.5** | **75.3** | **47.6** | **87.9** | **78.7** |

**0.154** on GenEval++ and **63.27** on DPG-Bench: comparable to SD v1.5 (63.18) on DPG-Bench, and above SDXL (0.061) but below larger unified models such as Janus-Pro (0.246). Both benchmarks are hard; even SD3-medium reaches only 0.296 on GenEval++, so a gap to top-tier generators is expected at our 256×256, low-cost setting. The injected generation pathway still follows compositional instructions to a reasonable degree, in line with our resource-efficient goal.

**(2)Qualitative Evaluation** Figure 4 illustrates the learning process of the model during the second stage of WoHG training. As the number of training epochs increases, there is a rapid improvement in both the fidelity of the images and their alignment with the text. This intuitively demonstrates that our framework can efficiently learn new generative capabilities without causing catastrophic forgetting (in conjunction with the quantitative analysis), thus validating the effectiveness of our "upgrade" paradigm. Figure 5 then showcases the model's output capabilities after full training. The model can autonomously select the response modality based on the instruction: (1) for a pure generation instruction (e.g., "draw an apple"), the model directly returns a single image; (2) for a complex request requiring both explanation and illustration, it can output a piece of text followed by a matching image. These examples indicate that ARMOR has successfully mastered modality selection and interleaved combination, proving that our architectural design and data strategy have effectively solved the challenge of interleaved text-image generation.

In summary, the experimental results demonstrate that our proposed framework effectively solves the challenges of catastrophic forgetting and interleaved text-image generation. Furthermore, compared to other unified models, it requires significantly fewer resources. This validates the effectiveness of our core viewpoint: that "upgrading" rather than "rebuilding" existing expert models is a highly efficient paradigm.

Table 3: GenEval and FID results. Gen. = generation-only; NoILO. = unified models without interleaved output; ILO. = unified models with interleaved output. Training cost is reported in A100-day equivalents (H100 $\sim$ 2.5$\times$, H800 $\sim$ 2$\times$ A100).

| Type | Method | #Param | #Train Images | Train Cost (GPU days) | Image Res | GenEval ↑ | FID ↓ |
|------|--------|--------|---------------|-----------------------|-----------|-----------|-------|
| Gen. | LlamaGen | 0.8B | 60M | – | 256 | 0.32 | 8.69 |
| | LDM | 1.4B | 400M | – | 1024 | 0.37 | 12.64 |
| | Emu3-Gen | 8B | – | – | 512 | 0.54 | 19.30 |
| | SDXL | 7B | 2000M | – | 1024 | 0.55 | 9.55 |
| | SDv3 (d=24) | 2B | – | – | 1024 | 0.62 | – |
| | SDv2.1 | 0.9B | – | 8333/A100 | 768 | 0.50 | 26.96 |
| | SDv1.5 | 0.9B | 2000M | 6250/A100 | 512 | 0.43 | 9.62 |
| | DALL-E2 | 6.5B | 650M | 4166/A100 | 1024 | 0.52 | 10.39 |
| | PixArt-alpha | 0.6B | 25M | 753/A100 | 1024 | 0.48 | 7.32 |
| NoILO. | VILA-U | 7B | 15M | – | 384 | 0.42 | 7.69 |
| | Show-o | 1.3B | 36M | – | 512 | 0.53 | 9.24 |
| | D-DiT | 2B | 400M | – | 512 | 0.65 | – |
| | TokenFlow-XL | 14B | 60M | – | 384 | 0.55 | – |
| | SynerGen-VL | 2.4B | 667M | – | 512 | 0.61 | 7.65 |
| | Janus-Pro-7B | 7B | 72M | 3584/A100 | 384 | 0.80 | – |
| | Janus-Pro-1B | 1.5B | 72M | 1568/A100 | 384 | 0.73 | – |
| | SEED-X | 17B | 158M | $\sim$960/A100 | – | 0.49 | 14.99 |
| | MetaMorph | 8B | 8M | – | – | – | 11.8 |
| ILO. | Chameleon | 7B | 1.4B | 35687/A100 | 512 | 0.39 | – |
| | Orthus | 7B | – | – | 512 | 0.58 | – |
| | **ARMOR** | **8B** | **5M** | **$\sim$500/A100** | **256** | **0.51** | **9.07** |

Table 4: Evaluation results on InterleavedBench Liu et al. (2024b). Scores are rated on a 1–5 scale by GPT-4o across five dimensions. TIC denotes Text-Image Coherence. Models above the line are end-to-end generators; models below are pipeline-based systems.

| Model | Text Quality | Perceptual Quality | Image Coherence | TIC | Helpfulness | AVG |
|-------|--------------|--------------------|-----------------|-----|-------------|-----|
| MiniGPT-5 | 1.22 | 2.45 | 1.62 | 2.03 | 1.77 | 1.82 |
| GILL | 0.75 | 3.21 | 2.25 | 1.53 | 1.48 | 1.84 |
| EMU-2 | 1.26 | 2.28 | 1.89 | 1.34 | 1.64 | 1.68 |
| EMU-2 (Gold Text) | 1.56 | 3.35 | 2.89 | 1.43 | 2.10 | 2.27 |
| **ARMOR** | 3.67 | 4.01 | 1.93 | 4.17 | 2.92 | 3.34 |
| Gemini1.5 + SDXL | **4.40** | 3.99 | **3.64** | 4.13 | 3.62 | 3.96 |
| GPT-4o + DALL · E 3 | 4.37 | **4.36** | 3.51 | **4.55** | **3.88** | **4.13** |

## 4.3 Ablation Studies

**(1)Capability Retention** To validate the contribution of our parameter freezing strategy in preventing catastrophic forgetting during training, we conducted an ablation study on the freezing module in the first training step. Specifically, we divide all potentially trainable parameters (excluding the vision tower and its connector) into six logical groups and fine-tune various combinations of these groups. Each experimental run uses the same data mixture (100k text-to-text, 300k text-image-to-text, 100k text-to-image, and 100k text-to-text-image) and hyperparameters, with performance evaluated on the MMMU, MME, and MMB benchmarks. The results in Table 7 reveal a clear pattern. Unfreezing core components of the original MLLM, such as the text embedding or decoder layers, leads to a significant drop in accuracy. In contrast, updating only our lightweight, additive components has a minimal impact. This confirms that our selective freezing strategy is critical for learning new generative skills without causing catastrophic forgetting of the model's foundational comprehension abilities.

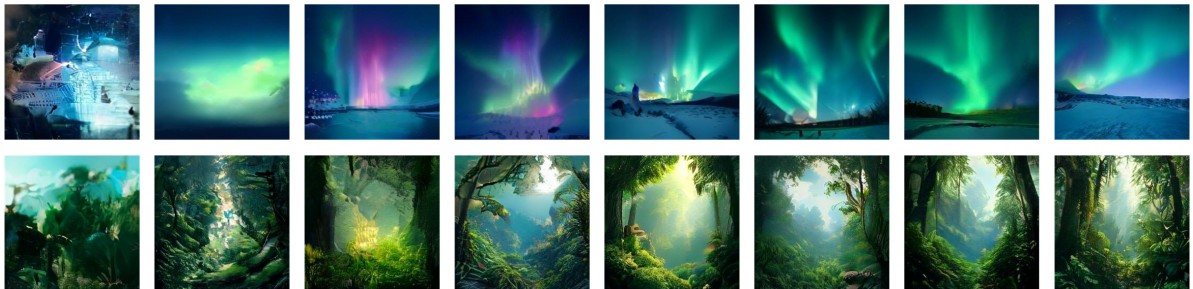

Figure 4: Demonstration of the optimization process of model generation effect in the second stage of WOHG. Prompt (upper part): Could you generate an image of the aurora for me?; Prompt (lower part): Please help me draw a picture of the tropical rainforest.

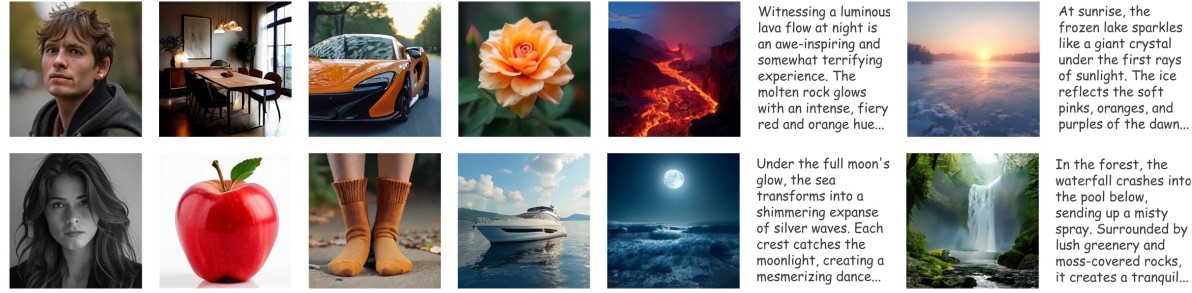

Figure 5: Example of hybrid modal output: text generates image (left half); text generates interleaved text and image (right half)

**(2)Experiment on Forward-switching Mechanism**

To test the impact of separating the text and image vocabularies, we train two variants for five epochs on 0.5 M samples (300 k t2i and 200 k t2ti) with the same hyper-parameters as before: (1) full ARMOR with the forward-switch, (2) an otherwise identical model that uses a single, shared output head. Figure 6a gives a qualitative comparison on the prompt "Can you help me draw a picture of a teddy bear doll?". Images (a–c) come from the single-head baseline: the first epoch already fails because the model emits an incorrect number of tokens, and later epochs still mix text and image codes, producing distorted bears. In contrast, images (d–f) are generated by the forward-switch model after epochs 1, 2, and 5. It produces valid pictures after only one epoch and continues to refine quality thereafter. Figure 6b shows the loss function curves under the two settings. Both visual and loss curve evidence confirm that the forward-switching mechanism eliminates the long-tail conflict between 50,000 text tokens and 8192 image encodings, thereby achieving faster and more stable convergence.

**(3)Experiment on Model Scale** To verify whether the method I proposed has scaling ability, we varied the adapter depth to 2 layers (2a) and 4 layers (4a). Both variants were trained on 0.3 M t2i samples (the Stage-3 dataset) and evaluated throughout training with GenEval and FID (COCO-30k). We considered models trained with Stage 1 only (S1) and with Stage 1 + Stage 2 (S2), using the same hyper-parameters as Stage 3 in in the previous text. Table 9 reports the results. The 2a model learns faster at the beginning, but the larger 4a model ultimately achieves better quality (lower FID and higher GenEval). GenEval shows the same trend. In addition, S2 consistently outperforms S1, confirming the benefit of the "how-to-generate" stage. Overall, the experiment indicates preliminary evidence that ARMOR benefits from increased adapter capacity.

**(4)Generality Across Backbones** To examine whether the ARMOR "upgrade" paradigm is tied to a specific base model, we apply the identical framework and WoHG training recipe to a stronger and more recent backbone, InternVL3-8B, without any architecture- or hyper-parameter-specific tuning. Table 8 reports, for each backbone, the understanding performance before and after the upgrade together with the

Table 5: Evaluation results for interleaved generation capabilities, including Switch-Accuracy and Interleave-CLIPScore.

| Model | Text Acc.(%) | Image Acc.(%) | Mixed Acc.(%) | CLIPScore |
|---|---|---|---|---|
| Anole | 55.3 | 46.1 | **92.4** | 19.4 |
| VARGPT | **96.1** | 0 | 17.9 | 30.8 |
| **ARMOR** | 94.9 | **88.7** | 91.5 | **35.6** |

Table 6: Compositional and dense-prompt text-to-image evaluation. GenEval++ (Ye et al., 2025b) (higher is better, max 1.0) and DPG-Bench (Hu et al., 2024) (higher is better, max 100) stress compositional attribute binding, counting, spatial relations, and long dense prompts. "–" denotes a score not reported by the corresponding source.

| Model | Image Res | GenEval++ ↑ | DPG-Bench ↑ |
|---|---|---|---|
| SD v1.5 | 512 | – | 63.18 |
| Janus-Pro | 384 | 0.246 | – |
| SD3-medium | 1024 | 0.296 | – |
| SDXL | 1024 | 0.061 | 74.65 |
| **ARMOR (ours)** | 256 | **0.154** | **63.27** |

Table 7: Ablation study on the impact of different trainable parameter combinations on multimodal understanding after Stage 1. Each experiment (Exp.) fine-tunes a different set of components, where ✓ denotes a trained component and × denotes a frozen one. The gray-highlighted row (Exp. 5) represents our proposed method, whose full three-stage results are reported in Tables 2–5. "Switch" represents the correct selection of the response modality; Exp. 6–7 lack this capability and thus cannot generate images. Since this ablation isolates Stage 1 ("what to generate"), where the image branch has not yet been trained, image-generation metrics such as GenEval/FID are not yet defined here; this table therefore deliberately measures only how training different modules interferes with understanding, while "Switch" captures the relevant Stage-1 capability. The full understanding–generation trade-off of our final three-stage model is reported in Tables 2 and 3.

| Exp. | Training Parameters | | | | | | Switch | Result | | |
|---|---|---|---|---|---|---|---|---|---|---|
| | InLM emb. | Add. emb. | Dec. layer | Txt out. | Adapter | Vis. out. | | MMMU | MME | MMB |
| 1 | ✓ | ✓ | ✓ | ✓ | ✓ | ✓ | ✓ | 36.8 | 1532.3 | 57.5 |
| 2 | × | ✓ | ✓ | ✓ | ✓ | ✓ | ✓ | 38.9 | 1617.4 | 65.3 |
| 3 | ✓ | ✓ | × | ✓ | ✓ | ✓ | ✓ | 35.6 | 1497.7 | 65.1 |
| 4 | × | ✓ | × | ✓ | ✓ | ✓ | ✓ | 50.9 | 2273.7 | 78.8 |
| **5 (Ours)** | × | ✓ | × | ✓ | × | × | ✓ | 51.5 | 2281.1 | 78.5 |
| 6 | × | × | × | ✓ | × | × | × | 52.2 | 2220.5 | 77.9 |
| 7 | × | ✓ | × | × | × | × | × | 53.1 | 2339.4 | 81.7 |

resulting generation quality. On InternVL3-8B, ARMOR preserves on average about 95% of the base model's understanding accuracy across the six benchmarks, closely mirroring the retention observed on InternVL2.5-8B (about 96%), while simultaneously acquiring competitive generation (GenEval 0.53, comparable to 0.51 on InternVL2.5). The fact that the same recipe both preserves understanding and endows generation on a markedly stronger backbone indicates that our core contribution does not hinge on a particular base model. We note that InternVL3 still belongs to the InternVL family, and we leave validation on other backbone families as future work.

Table 8: Generality of the ARMOR upgrade paradigm across backbones. "Ret." denotes the average percentage of the base model's understanding performance retained after the upgrade, computed over the six understanding benchmarks; "GenEval" measures text-to-image generation (base models cannot generate, denoted "–").

| Model | MMMU | MME | MMVet | MMB | Hall | POPE | GenEval | Ret.(%) |
|---|---|---|---|---|---|---|---|---|
| InternVL2.5-8B (base) | 53.5 | 2338.9 | 59.6 | 82.0 | 49.0 | 88.9 | – | – |
| ARMOR (InternVL2.5-8B) | 51.5 | 2281.5 | 56.3 | 78.5 | 47.6 | 87.9 | 0.51 | 96.7 |
| InternVL3-8B (base) | 72.6 | 2415.0 | 81.3 | 89.0 | 59.1 | 90.3 | – | – |
| ARMOR (InternVL3-8B) | 68.5 | 2337.4 | 77.4 | 83.6 | 55.7 | 86.2 | 0.53 | 95.0 |

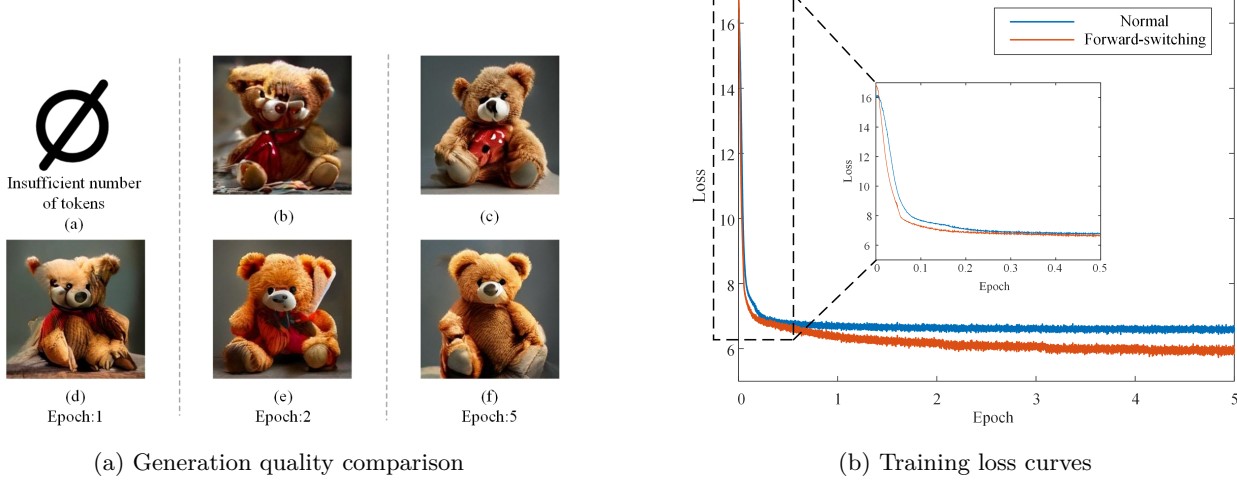

(a) Generation quality comparison  (b) Training loss curves

Figure 6: Ablation study on the forward-switching mechanism. (a) Single-head output (upper) vs. dual-head output (lower) during training. (b) Loss curves under both settings.

Table 9: Results of the Ablation Experiment on Model Scale.

| Result | Exp. | Mod. | Param | Training Progress | | | |
|---|---|---|---|---|---|---|---|
| | | | | 10% | 30% | 50% | 100% |
| FID↓ | S1 | 2a | 0.5B | 57.67 | 37.11 | 24.26 | 19.35 |
| | S1 | 4a | 0.9B | 65.52 | 42.77 | 26.67 | 16.33 |
| | S2 | 2a | 0.5B | 25.01 | 17.32 | 12.19 | 9.59 |
| | S2 | 4a | 0.9B | **19.31** | **14.20** | **11.08** | **9.07** |
| Gen. ↑ | S1 | 2a | 0.5B | 0.15 | 0.21 | 0.26 | 0.28 |
| | S1 | 4a | 0.9B | 0.13 | 0.19 | 0.26 | 0.31 |
| | S2 | 2a | 0.5B | 0.33 | 0.39 | 0.43 | 0.47 |
| | S2 | 4a | 0.9B | **0.35** | **0.42** | **0.46** | **0.51** |

## 5   Conclusion

In this paper, to address the issues of high training costs, weak understanding capabilities, and difficulty in supporting interleaved text-image output of UniMs, we propose the ARMOR framework to construct UniMs by upgrading existing MLLMs. By introducing an asymmetric encoder-decoder architecture and a forward switching mechanism, MLLMs can output naturally interleaved text and images. We collected high-quality interleaved datasets and developed a three-stage training algorithm called WoHG for fine-tuning

existing MLLMs. This algorithm enables MLLMs to possess unified capabilities in both understanding and generation tasks. Benchmark tests across multiple dimensions and numerous ablation results show that our framework can effectively endow existing MLLMs with generation capabilities while well preserving their understanding capabilities, making it simple and effective to upgrade MLLMs to UniMs.

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

# A    Implementation Details and Hyperparameters

This section provides a comprehensive overview of the experimental setup to ensure full reproducibility of our results.

**Hardware and Software.**    All experiments were conducted on a cluster of 8 NVIDIA H100 GPUs, each with 80GB of VRAM. Our implementation is based on PyTorch (version 2.6.0) and leverages the Transformers library (version 4.46.3) for model components. The VQ-VAE decoder is adapted from the official Chameleon implementation.

**Base Model and VQ-VAE.**    Our ARMOR model is built upon the publicly available **InternVL-Chat-V1.5** model, which has approximately 8B parameters. We integrate a pre-trained VQ-VAE decoder, originally from Chameleon, with a codebook size of 8192. The image decoder is responsible for translating the discrete image tokens back into $256 \times 256$ pixel space.

**Training Hyperparameters.**    The three-stage WoHG training algorithm employs specific hyperparameter configurations for each stage, as detailed in Table 10. The total training process for our final ARMOR model required approximately **500 A100-equivalent GPU-days**, as reported in the main paper.

Table 10: Detailed hyperparameter settings for each stage of the WoHG training algorithm. These settings were used for our main ARMOR model.

| Hyperparameter | Stage 1: What to Generate | Stage 2: How to Generate | Stage 3: How to Answer Better |
|---|---|---|---|
| *Dataset Configuration* | | | |
| t2t Data Size | 100k | - | 50k |
| ti2t Data Size | 300k | - | 300k |
| t2i Data Size | 100k | 2.5M | 300k |
| t2ti Data Size | 100k | 2.5M | 50k |
| *Optimizer and Scheduler* | | | |
| Optimizer | AdamW | AdamW | AdamW |
| Learning Rate | 4e-5 | 1e-4 | 5e-5 |
| $\beta_1, \beta_2$ | 0.9, 0.999 | 0.9, 0.999 | 0.9, 0.999 |
| Weight Decay | 0.05 | 0.05 | 0.05 |
| LR Scheduler | Cosine Annealing | Cosine Annealing | Cosine Annealing |
| Warmup Steps | 500 | 1000 | 200 |
| Gradient Clipping | 1.0 | 1.0 | 1.0 |
| *Loss Configuration* | | | |
| Text Loss Weight ($\alpha$) | 1.0 | 0.0 | 1.0 |
| Image Loss Weight ($\beta$) | 0.0 | 1.0 | 1.0 |

# B    Limitations and Future Work

We list below the main constraints of the current ARMOR prototype and how we plan to address them.

**Image resolution.**    All experiments use a $256 \times 256$ VQ-VAE for quick training and fair comparison with most autoregressive baselines. The architecture itself is resolution-agnostic: the image decoder can be replaced with any higher-resolution VQ or latent diffusion model by enlarging the image-token vocabulary, without touching the frozen MLLM or the forward-switch logic. As future work we will provide $384 \times 384$ and $512 \times 512$ checkpoints.

**On the visual encoder and the scope of generation.**    A natural question is whether the frozen, understanding-oriented backbone retains enough information for image generation. Our experiments answer this directly: ARMOR reaches a GenEval score of 0.51 and an FID of 9.07, producing good-quality images. Notably, although ARMOR reuses the image decoder of Chameleon, it clearly surpasses Chameleon itself on GenEval (0.51 vs. 0.39), which indicates that the frozen backbone already carries sufficient semantic

information to drive effective generation. Whether a stronger or more generation-specific visual encoder-or unfreezing and jointly training the encoder-could further improve generation quality is beyond the core scope of this paper. We do not deny that such choices may bring additional gains, but they would substantially increase the training cost and thus run counter to our central motivation: endowing a multimodal under-standing model with generation capability in a lightweight, non-invasive manner while preserving its original understanding.

**Single-image replies.** ARMOR currently generates at most one image per answer. However, its architecture natively supports interleaved output of multiple text-image segments. To supporting multiple images only requires allowing several `<imgbos>`–`<imgend>` segments in the same sequence; no architectural change is needed, but additional t2tii training data will be collected, we plan to construct a new training set of approximately 500k text-image interleaved examples. This data will be curated from public web documents and synthesized by prompting advanced multimodal models like GPT-4V, followed by rigorous filtering for quality and safety.

**Domain bias and safety.** The 5 M training images are filtered by CLIP-aesthetic and NSFW detectors but may still contain residual biases. We plan to integrate more rigorous safety filters and offer a community feedback channel for problematic generations.

**Resource footprint.** In terms of resource consumption, ARMOR is far lighter than existing UniMs, a full $512 \times 512$ model will still require additional GPU hours. We are exploring parameter-efficient finetuning (LoRA on adapters) and quantised decoding to further reduce cost.

**Broader Impact.** ARMOR enables existing MLLMs to generate visual content, which introduces potential risks such as the creation of misleading imagery. Since the MLLM backbone is kept entirely frozen, its original safety alignment is preserved. Nevertheless, the newly added generation pathway may produce biased or undesirable outputs. We encourage future work to integrate dedicated safety mechanisms for the generative components.

These limitations do not undermine the central claim-that a frozen MLLM can be upgraded to a unified model with minimal overhead, and they highlight promising directions for extending ARMOR's practicality and robustness.

## C Architecture Comparison

Figure 7 positions ARMOR against four representative unified models. (1) Chameleon: pure autoregressive (AR); trains one large vocabulary for both text and image tokens. (2) Show-o: AR text plus a separate diffusion decoder; high quality but high cost. (3) VILA-U: AR with thousands of RO tokens; large soft-max slows training. (4) Janus-Pro: two distinct vision branches (understanding / generation) trained from scratch. (5) ARMOR (ours): keeps the frozen MLLM encoder for understanding, adds a lightweight VQ decoder for generation, and uses a forward-switch to avoid vocabulary clash-only 0.9 B extra params while retaining 95 % of the base model's accuracy.

## D Dataset Examples

**1. `t2t` (Text-to-Text) Example:**

**From:** human

**Value:** What kind of scenery can be found beside the Nile River?

**From:** gpt

**Value:** The Nile River offers desert landscapes, lush farmlands, palm trees, small villages, ancient monuments, wildlife, and the fertile Nile Delta, creating a mix of natural beauty and historical significance.

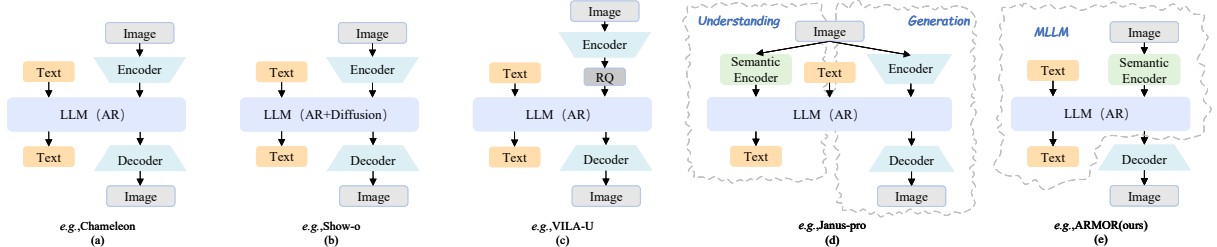

Figure 7: Architectural taxonomy. ARMOR inherits the simplicity of pure AR models (a) while being far lighter than hybrids (b–d).

**2. t2ti (Text-to-Text-Image) Example:**

**From:** human

**Value:** What kind of scenery can be found beside the Nile River? Can you draw an image to depict it for me?

**From:** gpt

**Value:** Alongside the Nile River, you can find lush, verdant banks filled with vibrant greenery and towering papyrus plants. The scene reflects a serene environment with occasional traditional felucca sailboats gliding across the water.

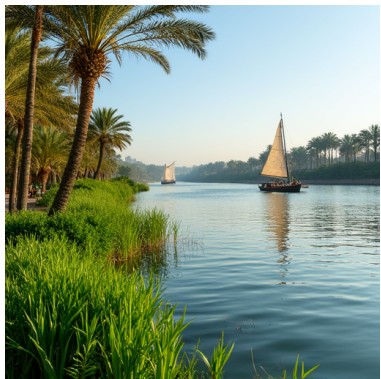

**3. t2i (Text-to-Image) Example:**

**From:** human

**Value:** Please draw an image of a sunset over the sea for me.

**From:** gpt

**Value:**

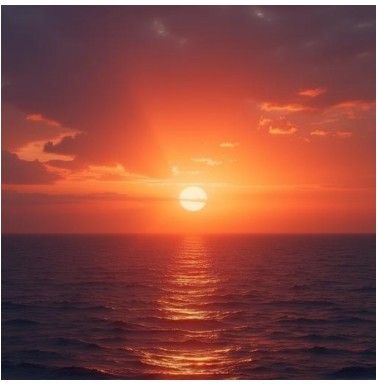

**4. ti2t (Text-Image-to-Text) Example:**

**From:** human

**Value:** <image>What is this photo about?

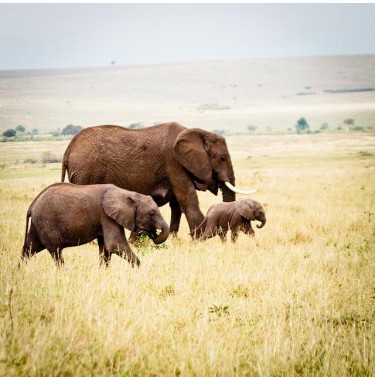

**From:** gpt

**Value:** In the heart of a vast grassland, a family of three elephants is on a journey. The adult elephant, with its majestic brown skin, leads the way in the center, its size significantly larger than the two smaller ones. On either side of this gentle giant, slightly behind, are two smaller elephants, their youthful energy evident in their stride. The elephants are making their way through a field of tall grass, their brown bodies contrasting beautifully with the greenery around them. The grassland extends far into the distance, dotted here and there with a few trees standing tall and solitary. Above them, the sky is a canvas of overcast clouds, casting a soft light over the scene. The relative positions of the elephants suggest a protective formation, with the adult elephant guiding its young ones through the grassland. This image captures a moment of tranquility and familial bond in the wild.

## E Data Ratio in the Training Phase

**In the first stage of training**, the experiment was set to train the newly added embedding layer and the output layer of internvl. Since training the output layer of the internvl model is bound to affect the original capabilities of the model, how to reduce the degree of this negative impact has become one of the key issues we need to solve. We tested and found that using the training scheme of t2t (100K), ti2t (100K), t2i (100K) and t2ti (100K) still reduces the original capabilities of the model, resulting in a worse instruction - following ability when the model faces some questions. Therefore, we carefully selected 300K data from the open - source dataset (mainly LLaVA - NEXT) for model training, so that the degree of decline in the original capabilities of the model after the first - stage training becomes very small.

**In the second stage of training**, the experiment was set to only train the newly added parts. Therefore, we used a large amount of t2i and t2ti data to train the generation ability of the model without having to worry about the impact of training new capabilities on the original capabilities.

**In the third stage of training**, in the experiment, the newly added embedding layer, the newly added adapter, the output layer of internvl, and the final visual output layer were set to be trained. Initially, the data ratio we used was t2t(50K), ti2t (50K), t2i (50K) and t2ti (50K). Due to the low - quality t2ti data further reducing the original ability level of the model, and the small scale of the image fine - tuning data volume, the final geneval ability score of the model was only 0.37. After we expanded t2i to (300k), we found that its geneval score reached 0.47. Furthermore, when we increased the newly added transformer layers of the model to 4 layers, we obtained a generation result of 0.51, and we believe there is still room for improvement. In addition, when we also expanded the ti2t data to 300k, compared with the initial training scheme, the MMMU score increased from 49.8 to 51.5, and there were also improvements to varying degrees in other dimensions of the benchmark. It is worth mentioning that for the ti2ti dialogue pattern, we did not specifically integrate this ability for the model. However, due to the characteristics of ARMOR, if there is a demand for image generation in the question, then ARMOR can naturally predict relevant image information based on the content of the text answer. We recorded the dialogue examples of ti2ti in the supplementary materials later.

## F   Work motivation and details

The purpose of ARMOR is to transform a pre-trained MLLM into a model with unified understanding and generation capabilities. Currently, unified understanding and generation models often require a large amount of data to train the model's multimodal understanding abilities. Some models attempt to achieve this function by using an external attachment approach, such as Metamorph and SEED-X. These two models are also implemented based on pre-trained models. However, during the training process, they rely on the original capabilities of the model for training. Although this can reduce the amount of training, it undoubtedly still damages the original capabilities of the model. ARMOR completely "stands on the shoulders of giants" to expand its capabilities. During the training process, we found that it is extremely difficult to retain the original capabilities of the model. Even when we only train a small part of the model's parameters, it can cause catastrophic damage to the model's capabilities. To this end, we have made a lot of efforts to retain the original capabilities of the model. In the first stage of training in this work, our initial goal was simple, which was to make the model learn to correctly generate the switch of the image modality (`<imgbos>`) according to the question. However, although the output of the model is normal after training, its performance on the benchmark almost collapses. At the same time, the inability to obtain all the fine-tuning data of the model is a challenge we face. Therefore, we have screened from open-source datasets and used the distillation method to obtain a batch of high-quality image-text understanding datasets. This can retain the original capabilities of the model to the greatest extent, but it will still cause some loss of capabilities. In future research, we will continue to explore more effective methods to make up for the current shortcomings.

## G   Mixed-modality Output Test

In this subsection, we list some examples of ARMOR's interleaved input and output, as well as comparisons with other models.

### G.1   Text-Image-to-Text-Image Test

**Question:** Does the picture show the scene of athletes exercising? Can you help me draw one similar to this picture?

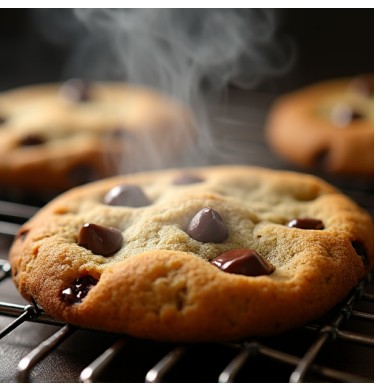

**ARMOR:** No, the picture does not show athletes in action. It depicts freshly baked chocolate chip cookies cooling on a wire rack.

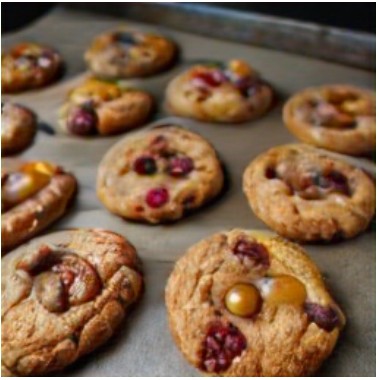

**Janus-Pro:** The picture you provided shows freshly baked chocolate chip cookies cooling on a wire rack. There are no athletes or exercise scenes in this image. I can't help you draw one similar to this picture as I am an AI language assistant and not capable of visual creation. If you have any other questions or need assistance with a different topic, feel free to ask!

**VILA-U:** No, the picture does not show the scene of athletes exercising. It is a close-up image of a freshly baked cookie on a cooling rack. The cookie is round in shape and has chocolate chips on top.

**Show-o: Yes, the image shows a scene of athletes exercising, with a person holding a pair of scissors** and a doughnut with chocolate frosting and sprinkles on a cooling rack. The doughnut is placed in the foreground, while the athlete is in the background. The athlete is holding the scissors, possibly preparing to cut the doughnut. The scene is set against a black background, which emphasizes the focus on the athlete and the doughnut.

**Anole: Gentle climate, nature-friendly outdoor activities, or simply cozy home comforts - the image triggers a sense of wellbeing in viewers. A partially visible sports coach calmly leads his group as they work through a warm-up exercise on what looks like a quaint sports field.**Freshly baked donuts lay ready on the grill, tempting the trained bods to break their routines with a sweet treat. The scene is both focused and relaxed, with each athlete having space to move and enjoy the time spent together. Here's an image based on the provided picture:

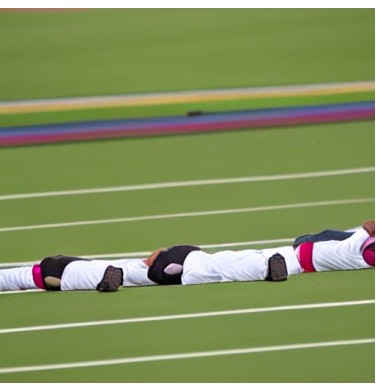

**VARGPT:** No, the picture does not show athletes exercising. It shows a close-up of a chocolate chip cookie on a cooling rack.

**Summary:** This is a evaluation integrating understanding and generation requirements. Through the evaluation, Janus-Pro can recognize that the image doesn't show athletes but a cookie, and responds to the subsequent image-generation request by clearly stating it's a language model unable to perform visual generation. VILA-U can also identify that it's not an athlete scene, yet makes no response to the image-generation demand. However, Show-o fails to accurately understand the image content. It even describes a false image caption in a seemingly plausible way and doesn't respond to the image-generation request at all. Compared with Janus-Pro, VILA-U and Show-o, ARMOR can accurately understand the image content, confirm it's not about athletes, and precisely recognize the image-generation requirement, generating an image that meets the needs.Although Anloe stated in the latter part of the answer that these were freshly baked cookies, in the first part of the answer, it was still describing scenes of athletes, and the image drawn was also related to athletes, indicating its deficiency in multimodal understanding ability. VARGPT can correctly identify the confusing question and provide a correct answer, but the probability of it generating a picture is too low. After 20 rounds of testing, it still didn't generate any images, indicating that it automatically ignored the drawing question in the title: Can you help me draw one similar to this picture?

### G.2 Text-to-Text-Image Test

**Question:** What's the Northern Hemisphere like in June? Can you draw an image about it for me?

**ARMOR:** In the Northern Hemisphere during June, it's typically summer with warmer temperatures and longer daylight hours. Regions closer to the poles experience the midnight sun, where the sun remains visible at midnight due to the region's tilt towards the sun. Plants are lush, and wildlife is active

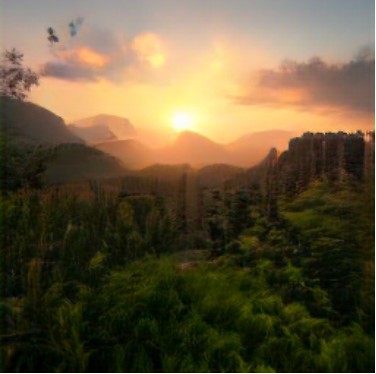

**Janus-Pro:** In the Northern Hemisphere during June, it's typically summer with warmer temperatures and longer daylight hours. Regions closer to the poles experience the midnight sun, where the sun remains visible at midnight due to the region's tilt towards the sun. Plants are lush, and wildlife is active

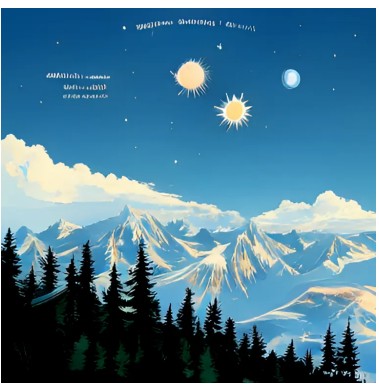

**VILA-U:**

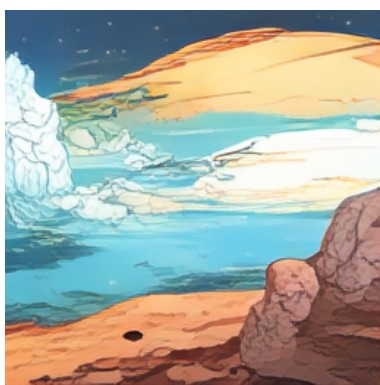

**Show-o:**

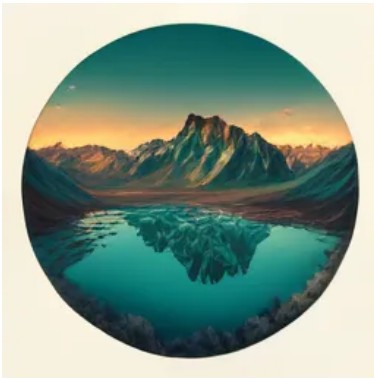

**Anole:** June is a beautiful time to visit the Northern Hemisphere. As the spring weather continues to improve, the region is transformed into a lush, vibrant landscape teeming with life. In the United States, Canada, and Mexico, the sun is shining brightly and temperatures are starting to rise. The days are long and warm, making it the perfect season for outdoor activities such as hiking, camping, and exploring national parks. One of the most noticeable changes in the Northern Hemisphere in June is the emergence of new vegetation. As the snow and ice begin to melt, plants and trees start to sprout up, filling the air with their sweet scent and vibrant colors. In the United States, for example, the blooming of wildflowers is a particularly stunning sight, with fields of purple lupines, yellow black-eyed Susans, and pink wild roses adding a pop of color to the landscape. Another significant change in the Northern Hemisphere in June is the return of wildlife. Many animals, such as birds, deer, and bears, emerge from their winter dens and make their way back to their habitats. In the United States, for example, birdwatchers can enjoy the spectacle of migrating flocks of geese, ducks, and pelicans flying overhead, while deer can be spotted grazing in fields and forests. In addition to the return of wildlife, June also marks the beginning of the summer season in the Northern

Hemisphere. This means that many popular tourist destinations are now open and ready for visitors. In the United States, for example, popular destinations such as beaches, theme parks, and national parks are all in full swing, providing opportunities for outdoor activities, relaxation, and cultural experiences. Overall, June is a wonderful time to visit the Northern Hemisphere. The warm weather, vibrant landscapes, and return of wildlife make it a season of renewal and growth, perfect for exploring the outdoors and experiencing the beauty of the natural world.

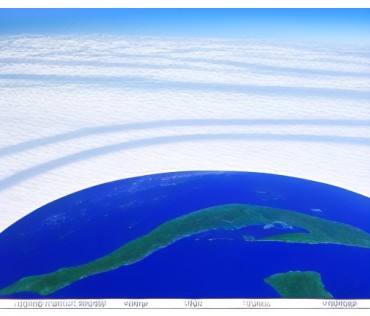

**VARGPT:** I created a drawing of the Northern Hemisphere in June. chatGPT.mm.

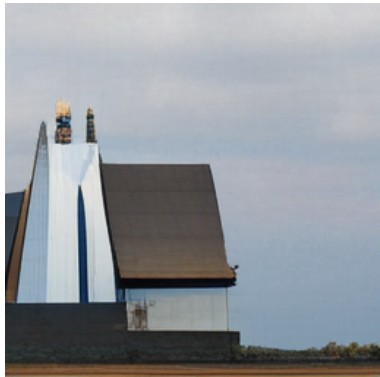

**Summary:** This evaluation focuses on multimodal generation tasks that require both text descriptions and corresponding images to fully address user requests. Unlike standard image generation, successful responses must provide accurate textual explanations paired with visually appropriate images.

Let's take a look at the output of these models. Janus - Pro, VILA - U, and Show - o can only generate a single image, while ARMOR, VARGPT, and Anole can output responses with alternating text and images. Both Janus - Pro and VILA - U generated snowy mountain scenes. Since June usually represents summer, these depictions of snowy mountains may not meet people's expectations of common summer scenes in June. Additionally, neither of these two images presents real natural landscapes. The image of Janus - Pro adopts a retro illustration style, while the image of VILA - U uses a flat illustration style. Now, look at Show - o. It generated an image of natural scenery, but it misinterpreted the word "hemisphere". The image it drew is circular, with hardly any elements showing summer characteristics. Anole correctly described the scene in the Northern Hemisphere in June, but its output is too long - winded, and the information presented in the image is difficult to distinguish. Although VARGPT demonstrated the ability to output alternating text and images, its text answer has no relation to the question at all, and the image depicts a house, which also doesn't match the question. In contrast, the text output of ARMOR is of moderate length, and it can output an image that highly matches the text content.

**Question:** What's the Southern Hemisphere like in June? Can you draw an image about it for me?

**ARMOR:** In June, the Southern Hemisphere is experiencing winter, with varying conditions across different regions. High latitudes may see continuous daylight or polar night depending on their proximity to the poles.

Temperate zones will have cold days typical of winter, with possible snowfall in mountainous areas. Tropical regions may experience rainy seasons with warmer temperatures.

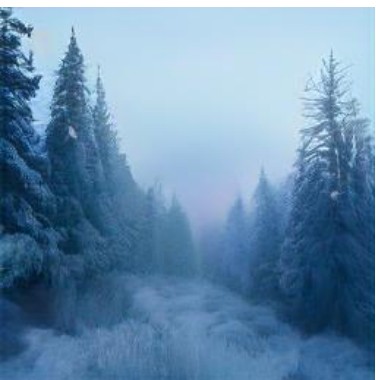

**Janus-Pro:**

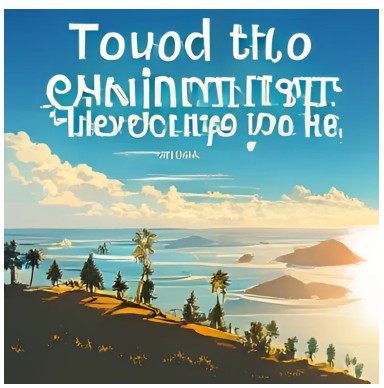

**VILA-U:**

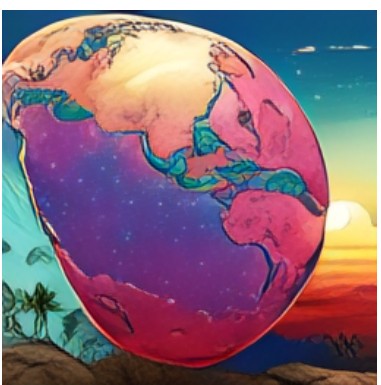

**Show-o:**

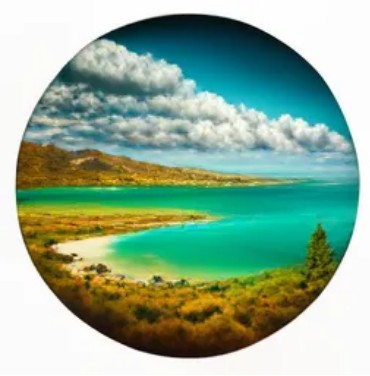

**Anole: In June, the southern hemisphere experiences a transitional season between spring and summer, with mild temperatures and changing weather patterns. The weather can vary widely depending on the specific location and time of year, but generally speaking, June is a pleasant month for exploring the southern hemisphere.** Below is a depiction of what the weather might be like in June for each location, using a color key to represent the average temperature for that area: * New Zealand (except the far south): Mild and pleasant, with average temperatures in the mid-10s Celsius (50s Fahrenheit). * Australia (except for the far north): Warm and sunny, with average temperatures in the mid-20s Celsius (70s Fahrenheit). * Southern Africa (excluding the far north): Warm and dry, with average temperatures in the mid-20s Celsius (70s Fahrenheit). * South America (except for the far north): Mild and pleasant, with average temperatures in the mid-10s Celsius (50s Fahrenheit). Keep in mind that these are general averages and weather patterns can vary significantly from year to year. It's always a good idea to check the current weather forecast before planning your trip.

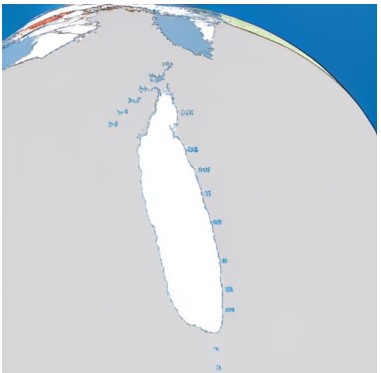

**VARGPT: I'm sorry, but I am not able to create a drawing of the Southern Hemisphere in June. chatGPT. I am a text-based AI model and do not have the capability to create visual images.** However, I can provide you with information about the Southern Hemisphere in June. In June, the Southern Hemisphere is in the midst of winter. The southernmost part of the world, including Antarctica, is experiencing the longest and darkest month of the year. The days are short, with the sun barely visible above the horizon, and the nights are long and cold. Temperatures in the Southern Hemisphere in June range from freezing in Antarctica to warm in the northern parts of South America and Africa. In general, the weather is cooler and drier in the southern parts of the hemisphere, while the northern parts are warmer and more humid. The Southern Hemisphere is also home to several summer festivals and events in June, including the Sydney Festival in Australia, the Stellenbosch Chenin Blanc Festival in South Africa, and the Calla Lily Festival in Chile.

**Summary:** This question simply replaces the Northern Hemisphere in the previous example with the Southern Hemisphere. Through testing, neither Janus-Pro, VILA-U, nor Show-o generated the pictures that met the requirements. Although the works generated by Janus-Pro and Show-o are of good quality, they do not match the scene in the Southern Hemisphere in June. VILA-U tried to draw a globe, which not only

fails to meet the requirements of the question but also has poor generation quality. In addition, the picture drawn by Show-o is still circular. Anole was able to output content with interleaved images and text, but there was a common-sense error in the first part of its text answer, which still indirectly shows the problem of Anole's relatively weak comprehension ability. And VARGPT showed a phenomenon of refusing to answer before starting the text answer. After our 20 attempts, including modifying the prompt to make it more in line with VARGPT's way of asking questions (for example: Please help me draw a picture of the scenery in the Southern Hemisphere in June. Can you draw a picture of the scenery in the Southern Hemisphere in June for me? ...), it still failed to successfully output the picture. Finally, given its strong comprehension ability, ARMOR was able to successfully generate the text content corresponding to the question, and at the same time generate a picture that is highly relevant to the text.

| Epoch | Text_L. | Img. Emb. | Visual_L. | Adp. | Und. |
|---|---|---|---|---|---|
| 10 | ✓ | ✗ | ✓ | ✓ | 0.78 |
| 10 | ✓ | ✓ | ✓ | ✓ | 0.72 |
| 20 | ✓ | ✗ | ✓ | ✓ | 0.64 |
| 20 | ✓ | ✓ | ✓ | ✓ | 0.48 |

Table 11: A Single Train Stage Result of Understanding Ability. Text_L denotes the text output layer. Img. Emb. denotes the *newly added* image-token embedding (the original text embedding is *not* trained). Visual_L denotes the visual output layer. Adp. denotes the added transformers adapter. Und. denote the percentage of the original InternVL2.5 capabilities mantained after training. In all configurations the original text embedding and decoder layers are kept frozen; the trainable set therefore corresponds to a single-stage variant that jointly updates every newly added component together with the text output head.

| Stage | Text_L. | Img. Emb. | Visual_L. | Adp. | Und. |
|---|---|---|---|---|---|
| Stage1 | ✓ | ✓ | ✓ | ✓ | 0.97 |

Table 12: Train The First Stage with A Small Amount of Data.

## H  Ablation Study on Training Strategy: From Single-Stage to Multi-Stage

Our initial approach explored a single-stage, full fine-tuning strategy to endow the model with generation capabilities while preserving its original understanding. We tested several configurations, with the results summarized in Table 11. We emphasize that these configurations constitute exactly the natural single-stage baseline one would compare against: the original text embedding and decoder layers are frozen throughout, while *all* newly added components (image-token embedding, visual output layer, adapter) are trained jointly with the text output head on the full mixture of data and tasks in one stage. This baseline directly isolates the contribution of the WoHG curriculum against simply "training everything trainable at once".

The experiments revealed a critical trade-off. As shown in Table 11, any scheme involving training the original text output layer led to a severe degradation of the base model's understanding capabilities, with performance dropping to as low as 48% of its original level. We attribute this catastrophic forgetting to the high-volume generation data overwhelming the carefully calibrated weights of the original MLLM's language head. Crucially, this represents an unfavourable trade-off in both directions: the loss of understanding is not compensated by any gain in generation, since (as discussed below) the single-stage image branch fails to converge and its quality stagnates. A configuration that sacrifices the model's core understanding without delivering usable generation is therefore strictly dominated by the multi-stage protocol, whose final understanding–generation balance is reported in Tables 2 and 3.

This finding invalidated the single-stage approach and motivated our pivot to the multi-stage WoHG training algorithm. The core idea became to first teach the model *when* to generate (Stage 1) using a minimal, targeted dataset, thereby protecting the text output layer. This initial stage, detailed in Table 12, successfully equipped the model with modality switching capabilities while preserving 97% of its understanding performance.

With the model's understanding capabilities secured, we proceeded to Stage 2, focusing solely on image generation. Here, we evaluated two schemes: (a) training only the "visual output layer + adapter", and (b) training the "newly added embedding + visual output layer + adapter". We found that scheme (a) was insufficient; the model's loss plateaued at a high level, and image quality stagnated (see Figure 8 for examples). We hypothesize that without fine-tuning the image token embeddings, a robust semantic link between text and the visual vocabulary cannot be formed.

Therefore, the final decision to adopt scheme (b) for Stage 2 was not arbitrary but the outcome of a systematic elimination process. This entire methodological journey-from identifying the failures of single-stage training to the rigorous, data-driven selection of our final multi-stage protocol-underwent a final verification. Our internal checklist confirms that this process adheres to the principles of a Comprehensive Methodology, ensures Rigorous and Reproducible Evaluation, and forms the basis for what we present as a Clear Presentation of a Significant Contribution to resource-efficient unified model training. This foundational work lays the groundwork for the generation capabilities we subsequently demonstrate.

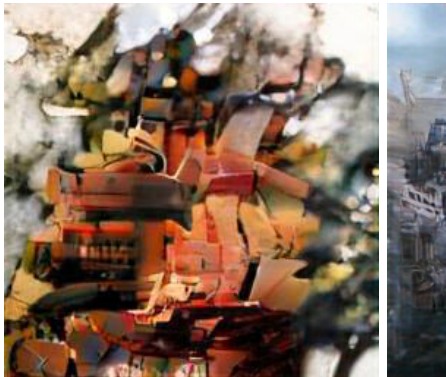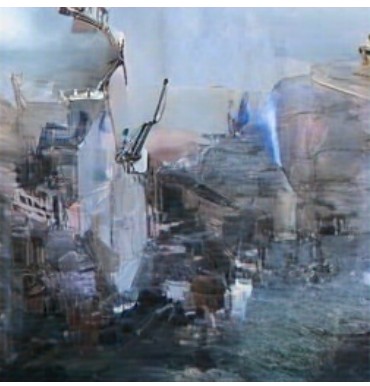

Figure 8: The Generated Images Trained with Visual Output Layer and Adapter.

