# OpenReview forum: "Empowering Multimodal Understanding Model with Interleaved Multimodal Generation Capability"
_TMLR — Accepted by TMLR_

### Review · Reviewer_R2sJ · 2025-11-24

**Summary Of Contributions:**

Summary

This paper introduces ARMOR, a resource-efficient and non-invasive framework that upgrades existing Multimodal Large Language Models into unified models capable of multimodal understanding and interleaved text–image generation. The method attaches a lightweight image decoder to a frozen MLLM backbone and employs a forward-switching mechanism using "imgbos/imgend" tokens to alternate between text and image output heads. To prevent catastrophic forgetting, the authors propose a three-stage curriculum (WoHG) that progressively teaches modality selection, image generation, and mixed-modality refinement. Experiments across nine understanding benchmarks, GenEval and FID for visual generation, and two new interleaved generation metrics show that ARMOR retains approximately 95% of the original MLLM’s understanding ability while achieving competitive generation quality at significantly reduced computational cost.

Strengths:

1. The paper presents a clear and timely problem: the difficulty of training Unified Models from scratch and the performance gap between UniMs and expert MLLMs. The proposed “upgrade rather than rebuild” paradigm is both practical and conceptually valuable.
2.  The asymmetric architecture and forward-switching mechanism are intuitive, well-motivated, and effectively avoid softmax long-tail competition between text and image tokens. The design is simple yet elegant, and Figures 1–2 clearly illustrate the workflow.
3.  The three-stage WoHG curriculum is an effective and well-structured solution to catastrophic forgetting, allowing the model to learn new generative skills while preserving the original understanding capabilities of the backbone.
4.  The experimental evaluation is comprehensive and convincing, covering multimodal understanding, standard image generation, and interleaved text–image generation. The grouping of baselines makes comparisons clear and highlights ARMOR’s positioning within the landscape of Unified Models.

**Audience:**

Yes

**Audience Explanation:**

The findings of this paper would be of clear interest to a substantial portion of TMLR’s audience. The work targets an active and rapidly evolving area at the intersection of multimodal large language models, unified understanding–generation models, parameter-efficient training, and catastrophic forgetting mitigation—all of which are topics that the TMLR community frequently engages with. The proposed “upgrade rather than rebuild” paradigm provides a practical and novel alternative to fully training Unified Models from scratch, addressing both computational efficiency and performance degradation issues that many researchers face in practice.

**Broader Impact Concerns:**

Overall, the submission does not raise major ethical concerns, but a brief Broader Impact statement would still be beneficial. Since ARMOR enables interleaved text–image generation and expands the capabilities of existing MLLMs, it would be helpful for the authors to briefly discuss potential considerations around safety and data bias. In particular, clarifying how the model handles inappropriate or low-quality generations, and acknowledging that training data may introduce residual biases, would strengthen the completeness of the paper. These points do not represent severe issues, but addressing them would make the broader impact of the work clearer to readers.

**Claims And Evidence:**

Yes

**Claims Explanation:**

Overall, the claims made in the submission are supported by accurate, convincing, and clearly presented evidence, though with a few areas that could benefit from additional analysis. The experimental section is comprehensive and evaluates the proposed ARMOR framework across three major dimensions—multimodal understanding, visual generation, and interleaved text–image generation—using widely recognized benchmarks such as MMMU, MME, MMVet, GenEval, and FID. The results consistently demonstrate that ARMOR retains approximately 95% of the base model’s understanding capability while significantly improving generation and interleaved output performance. The paper further includes extensive ablation studies (e.g., parameter freezing strategies) that substantiate the necessity of the proposed architectural choices and the WoHG curriculum.

**Requested Changes:**

1. The construction of the training datasets, especially the high-quality interleaved corpus, is described at a high level, and important details such as filtering thresholds and prompt generation rules are not fully documented, limiting reproducibility.
2. The motivation for the three-stage WoHG curriculum remains partly empirical. The main text does not sufficiently analyze whether alternative curricula or different stage boundaries were considered, leaving the design space somewhat unexplored.
3. The theoretical justification for the forward-switching mechanism is limited. The paper claims it eliminates vocabulary conflict and stabilizes convergence, but the explanation is mostly intuitive and lacks quantitative or analytical support.
4. The generated image quality, while competitive given the small decoder, is still noticeably behind specialized generative models. The paper could better clarify the intended positioning of ARMOR—not as a SOTA generator, but as a unified model balancing understanding and generation.

---

> ### Author Response · Authors · 2026-01-12
> **Rebuttal by Authors**
>
> We sincerely thank the reviewer for the thorough and professional evaluation. These suggestions have been invaluable in improving our work. Below we address each point in detail. A revised PDF will be uploaded shortly, with all modifications highlighted in blue.
>
> ---
>
> ### Response to Q1: Dataset Construction and Reproducibility
>
> We appreciate your query about reproducibility. Regarding the details of dataset construction, we provide the following clarifications:
>
> To ensure reproducibility, all training datasets used in this paper are **fully open-source** (links will be provided in the final version). As stated in Section 3.3, our interleaved dataset construction extends based on another work (InterSyn[1]). While the main text provided a high-level description, that work comprehensively elaborates the data cleaning pipeline and synthesis strategies. Nonetheless, we fully agree with the reviewer's suggestion. In the revised version, we will add detailed dataset information to the Appendix to ensure readers can clearly understand every step of the data construction process.
>
> ---
>
> ### Response to Q2: Motivation for the WoHG Curriculum
>
> We sincerely thank you for the suggestion. In fact, we have already conducted a detailed exploration of the training settings. As shown in Appendix H, we evaluated **nearly all possible combinations and analyzed the performance of each variant.** Experimental results demonstrate that the current WoHG strategy is indeed optimal in balancing understanding and generation capabilities. We note that this analysis content may have been overlooked due to formatting issues. In the revised version, we will add an explicit reference and a brief summary of Appendix H in the main text to ensure this key design justification is clearly visible to readers.
>
> ---
>
> ### Response to Q3: Theoretical Justification for the Forward-Switching Mechanism
>
> We appreciate your valuable feedback and provide the following explanations:
>
> 1. **Theoretical Justification:** The theoretical limitation of Shared Output Head lies in the **optimization conflict caused by global normalization.** Under a shared head setting, text prediction and image generation **are forced to compete within a single Global Softmax probability space, leading to suppression effects between the logits of different modalities.** Our forward-switching mechanism decouples the joint probability into two independent conditional distributions, $P(text|x)$ and $P(image|x)$. From an optimization perspective, this decoupling prevents mutual interference between the different layers, achieving Gradient Isolation, which fundamentally ensures training stability.
> 2. **Experimental Verification:** In the ablation study in Section 4.3, we compared the performance of the two architectures. The results clearly demonstrate the advantages of our method: Qualitatively, as shown in Figure 6, our method rapidly learns to generate complete images, whereas the shared head model exhibits generation collapse in the early stages. Quantitatively, regarding training loss (as shown in Figure 7), our method significantly outperforms the shared head baseline in both convergence speed and loss magnitude.
>
> To enhance clarity, we will supplement the revised version with this theoretical discussion and strengthen the relevant descriptions.
>
> ---
>
> ### Response to Q4: Positioning of Generation Quality
>
> We sincerely appreciate your precise summarization, which aligns with our core contribution. **ARMOR's value lies not in pursuing ultimate image fidelity, but in empowering the model with effective generative capabilities at a low cost while preserving its original understanding abilities.** Following your suggestion, we will include a detailed positioning clarification in the revised version.
>
> ---
>
> ### Broader Impact
>
> Thank you for the suggestion. We have added a Broader Impact paragraph in Appendix B, discussing potential considerations around safety and data bias, including how the frozen backbone preserves the original model's safety alignment and the acknowledgment that training data may introduce residual biases.
>
> ---
>
> We sincerely thank the reviewer for the professional and constructive feedback, which has been instrumental in strengthening the paper. We will upload the revised PDF shortly, with all modifications highlighted in blue for easy reference.
>
> [1] Feng, Yukang, et al. A High-Quality Dataset and Reliable Evaluation for Interleaved Image-Text Generation. arXiv, 2025, arXiv:2506.09427.
>
> Best regards,
>
> The Authors

---

> > ### Comment · Reviewer_R2sJ · 2026-06-03
> >
> > The authors have addressed my concerns.

---

> > > ### Author Response · Authors · 2026-06-03
> > > **Response to Reviewer's Comments**
> > >
> > > Dear Reviewer,
> > >
> > > We sincerely thank you for the time and effort you devoted to reviewing our manuscript, and for your constructive and professional comments. We are very glad that our responses have addressed your concerns.
> > >
> > > Best regards,
> > >
> > > The Authors

---

### Review · Reviewer_7WvV · 2025-12-18

**Summary Of Contributions:**

The paper proposes ARMOR, a framework designed to "upgrade" existing Multimodal Large Language Models (MLLMs)—specifically InternVL2.5-8B—into unified models capable of both understanding and interleaved text-image generation. The authors address two primary challenges in this domain: the high computational cost of training unified models from scratch and the issue of "catastrophic forgetting," where generation training degrades the model's original understanding capabilities.

The key contributions are:

Asymmetric Architecture: Integrating a lightweight, pre-trained VQ-GAN decoder (from Chameleon) with a frozen MLLM backbone via a lightweight adapter.

Forward-Switching Mechanism: A hard-gating strategy that separates the output heads for text and image tokens to avoid vocabulary conflict and long-tail distribution issues.

WoHG Training Strategy: A three-stage curriculum designed to introduce generation capabilities progressively.

Performance: The authors claim the model retains over 95% of the original understanding performance (tested on MMMU, MME, etc.) while achieving competitive generation performance (GenEval 0.51) at a fraction of the cost (500 A100 GPU days) compared to training from scratch.

**Additional Comments:**

NA

**Audience:**

Yes

**Audience Explanation:**

The topic of unified multimodal understanding and generation is active. The community is specifically interested in methods that reduce the massive computational barrier to entry for training these models. A methodology that allows "upgrading" existing open-source MLLMs without ruining their perception performance is of high practical value.

**Broader Impact Concerns:**

The authors have included a brief statement on safety in the appendix (filtering training data). However, since the model can generate images based on interleaved text context, it introduces new risks regarding the generation of misleading visual misinformation that is contextually coherent with a false narrative. A brief expansion on the implications of interleaved fabrication (fake news articles with consistent fake images) would be appropriate in the Broader Impact section.

**Claims And Evidence:**

Yes

**Claims Explanation:**

The authors provide extensive benchmarks for multimodal understanding, showing that ARMOR (InternVL2.5 based) maintains scores very close to the base InternVL2.5 model, supporting the claim of preventing catastrophic forgetting. They also provide ablation studies justifying the forward-switching mechanism and the freezing strategies. The training cost comparison supports the claim of efficiency.

**Requested Changes:**

There are several concerns

1. Comparison with "Pipeline" Baselines

The paper compares ARMOR primarily against other unified models (Janus-Pro, Show-o, Chameleon) and generation-only models (SDXL, etc.). However, a very strong and practical baseline for "multimodal understanding + generation" is simply cascading a sota understanding model (e.g., Qwen2.5-VL, GPT-4V) with a SOTA generation model (e.g., FLUX, SD3, DALL-E 3). From results provided in table 2&3, it seems that a user can achieve significantly higher image resolution and fidelity by prompting SD3 based on the output of Qwen2.5-VL. The paper claims unified models are needed for "interleaved" content, but does not forcefully demonstrate why ARMOR is superior to a pipeline in specific interleaved contexts. The paper should include a qualitative or quantitative comparison against a pipeline baseline (e.g., InternVL2.5 + SDXL/SD3). Show a specific failure case of the pipeline approach (e.g., consistency in multi-turn interleaved dialogue or shared context retention) that ARMOR solves. If the pipeline yields better images but lacks context sensitivity, this trade-off should be explicitly analyzed.

2. Image Resolution and Quality Limitations

The current implementation uses a 256x256 VQ-VAE decoder. 256px resolution is quite low and limits the practical utility of the generation (lowest in table 3). This also raise concern about if the comparison under GenEval/FID with different resolution is fair. The author should compare the image quality by up/down sampling to a same resolution.
While retraining with a larger decoder might be out of scope for the rebuttal, the authors should explicitly discuss the scalability of the VQ-VAE approach. Does the "Forward-Switching" mechanism hold up if the image token length increases significantly (e.g., for a higher res tokenizer)? A brief discussion in the "Limitations" section regarding the path to 1024px generation would be beneficial.

3. Clarification on the Forward-Switching Mechanism (Strengthen)

The forward-switching mechanism uses <imgbos> to switch heads. Please clarify how the training handles the non-differentiable nature of this switch if it is a hard decision during inference. During training, are both heads computing loss, or is it masked strictly based on ground truth tokens?

4. Data Efficiency Claims.

The paper claims "less than 1/70 the cost of training from scratch." Does "from scratch" include the pre-training of the vision encoder and the LLM backbone? Since ARMOR relies on a pre-trained heavy MLLM (InternVL), the cost comparison should be clearly labeled as "marginal cost to add generation" vs "cost to train a unified model from random initialization.

5. Figure 1 shows a "VQGAN", an obvious error.

---

> ### Author Response · Authors · 2026-01-12
> **Rebuttal by Authors**
>
> ## Part1
>
> We sincerely thank the reviewer for the thorough and professional evaluation. These suggestions have been invaluable in improving our work. Below we address each point in detail. A revised PDF will be uploaded shortly, with all modifications highlighted in blue.
>
> ---
>
> ### Response to Q1: Comparison with Pipeline Baselines
>
> Thank you for your valuable feedback. We acknowledge that the pipeline approach is indeed a strong baseline for high-resolution generation. However, we argue that unified models hold unique research value that cannot be replaced by pipelines. We would like to clarify the following points:
>
> 1. **Developing end-to-end unified models is a key trend in the research community.** Unified models like Show-o and Chameleon also trade off some generation fidelity compared to diffusion models (e.g., FLUX, SD3). ARMOR's core contribution lies in demonstrating that we can efficiently "upgrade" an existing MLLM into a unified model without training from scratch. This offers a much more accessible and resource-efficient path for the field compared to training unified models from the ground up.
> 2. **Information loss is a critical bottleneck in pipelines.** For the pipeline, the generation model relies on intermediate text descriptions, meaning it is "blind" to the original visual details. In contrast, ARMOR leverages the MLLM's powerful visual capabilities to process visual tokens, which are shared within the autoregressive generation context. This allows the model to **maintain superior semantic consistency during visual generation**, highlighting a fundamental advantage of unified models over pipelines.
> 3. **Quantitative evidence from InterleavedBench.** Following another reviewer's suggestion, we have evaluated ARMOR on InterleavedBench, an established benchmark that also includes pipeline-based systems. The results provide direct evidence for the above argument:
>
> | Model | Text Quality | Perceptual Quality | Image Coherence | TIC | Helpfulness | AVG |
> |-------|:------:|:------:|:------:|:------:|:------:|:------:|
> | MiniGPT-5 | 1.22 | 2.45 | 1.62 | 2.03 | 1.77 | 1.82 |
> | GILL | 0.75 | 3.21 | 2.25 | 1.53 | 1.48 | 1.84 |
> | EMU-2 | 1.26 | 2.28 | 1.89 | 1.34 | 1.64 | 1.68 |
> | **ARMOR** | **3.67** | **4.01** | **1.93** | **4.17** | **2.92** | **3.34** |
> | Gemini1.5+SDXL | 4.40 | 3.99 | 3.64 | 4.13 | 3.62 | 3.96 |
> | GPT-4o+DALL·E 3 | 4.37 | 4.36 | 3.51 | 4.55 | 3.88 | 4.13 |
>
> Notably, ARMOR's Text-Image Coherence (4.17) surpasses Gemini1.5+SDXL (4.13), while pipeline systems excel in Image Coherence due to their higher-resolution generators. This illustrates a clear trade-off: **pipelines offer better image fidelity, while unified models like ARMOR achieve tighter text-image semantic alignment.**
>
> ---
>
> ### Response to Q2: Image Resolution and Quality Limitations
> We sincerely appreciate your valuable and professional feedback. Regarding the concerns raised, we would like to make the following clarifications:
>
> 1. **Evaluation Fairness:** The GenEval metric focuses on evaluating coarse-grained semantic content, such as categories, colors, and positions. We believe that while the 256×256 resolution is limited in pixel details, **it is sufficient to cover this coarse-grained semantic information, allowing the detection model to recognize content and score effectively.** Furthermore, the FID evaluation inherently meets the reviewer's requirement, as the standard protocol mandates upsampling or downsampling images to a uniform resolution (299×299) prior to assessment.
> 2. **Scalability:** The forward-switching mechanism proposed is a hard gating logic based on special tokens, and its logical complexity is **independent of sequence length.** This is because the modal switching capability is learned during the first-stage training, where only the model's text output capability is trained, which is unrelated to the length of image tokens. Furthermore, under the current settings, the challenge of generating high-resolution images lies in the increase in sequence length, which is a universal issue faced by all autoregressive models. Currently, **there are mature solutions in academia,** such as the widely used multi-token prediction methods [1]. So this is not our bottleneck. In the future, we can support high-resolution generation by replacing the VQ-VAE with a higher compression rate or by extending the context window.
>
> Following your suggestion, we have added a supplementary discussion in the 'Limitations' section of the revised version.

---

> ### Author Response · Authors · 2026-04-12
> **Rebuttal by Authors**
>
> ## Part2
>
> ---
>
> ### Response to Q3: Forward-Switching Mechanism Clarification
> We appreciate your professional comment. Regarding this concern, we provide the following clarifications:
>
> 1. **Non-differentiability:** Our training process consists of three stages. The first stage aims to teach the model to select the correct response modality. **This process only involves training the model's text output capability (since `<imgbos>` is a text token).** Therefore, predicting this switch is an autoregressive classification task optimized via cross-entropy loss, which avoids any issues related to non-differentiability.
> 2. **Loss Computation:** The loss calculation is **strictly masked based on the Ground Truth.** As formally defined by the functions $\mathbb{I}_{text}(t)$ and $\mathbb{I}_{img}(t)$ in Equations (1) and (2): when the Ground Truth is a text token, we compute only the Cross-Entropy Loss for the text head, while the image head's output is masked (gradients are blocked); conversely, only the image head's loss is computed for image tokens. This design ensures that each head focuses exclusively on learning the prediction for its corresponding modality, as detailed in Section 3.3.
>
> To enhance clarity, we will strengthen the relevant descriptions in the revised version.
>
> ---
>
> ### Response to Q4: Data Efficiency Claims
>
> Thank you for pointing out this issue. We would like to clarify that the "1/70 cost" comparison refers specifically to **the marginal cost of adding generation capability to an existing pre-trained MLLM**, which is precisely the core positioning of ARMOR's "upgrade" paradigm. All unified models in our comparison, including Janus-Pro and Chameleon, also rely on pre-trained components (e.g., pre-trained LLM backbones and vision encoders)—this is standard practice in the field. Nonetheless, we agree that this should be stated more explicitly, and have revised the cost description in the manuscript to clearly label it as the marginal cost of capability expansion.
>
> ---
>
> ### Response to Q5: Figure 1 Error
>
> Thank you for pointing out this error. The label in Figure 1 should read "VQ-VAE Decoder" instead of "VQGAN." We have corrected this in the revised version.
>
> ---
>
> ### Broader Impact
>
> Thank you for raising this concern. We have expanded the Broader Impact discussion in Appendix B to address the risk of interleaved fabrication, alongside the existing discussion on safety filtering and backbone freezing for preserving safety alignment.
>
> ---
>
> We sincerely thank the reviewer for the professional and constructive feedback, which has been instrumental in strengthening the paper. We will upload the revised PDF shortly, with all modifications highlighted in blue for easy reference.
>
> [1] Gloeckle, Fabian, et al. "Better & Faster Large Language Models via Multi-token Prediction." ICML 2024, pp. 629:1-629:29, Vienna, Austria.
>
> Best regards,
>
> The Authors

---

### Review · Reviewer_y27a · 2026-04-05

**Summary Of Contributions:**

This paper proposes ARMOR, a framework that upgrades a pre-trained multimodal large language model (MLLM)—specifically InternVL2.5-8B—with image generation capabilities, rather than training a unified model from scratch. The framework introduces three main technical components: (1) **an asymmetric architecture** that attaches a lightweight image decoder (~0.9B parameters, borrowed from Chameleon's VQ-VAE) to the frozen MLLM backbone, (2) **a forward-switching mechanism** using special tokens to hard-gate between separate text and image output heads, enabling interleaved text-image generation, and (3) **a three-stage "What-or-How-to-Generate" (WoHG) training algorithm** that progressively teaches modality selection (Stage 1), image generation (Stage 2), and joint refinement (Stage 3) while carefully controlling which parameters are frozen at each stage.

The authors report that ARMOR retains over 95% of InternVL2.5's original understanding performance while achieving image generation at approximately 1/70 the cost of training from scratch.


**Key Strengths**

- The upgrade rather than retrain from scratch paradigm is practically motivated and addresses a real pain point: the prohibitive cost of training unified models from scratch.
- The WoHG training algorithm is well-designed with clear motivation for each stage. The progressive decomposition of learning objectives with corresponding parameter isolation is principled and well-articulated.
- The parameter freezing ablation (Table 5) systematically demonstrates which components are critical to preserve, clearly showing that unfreezing core MLLM components (original embeddings, decoder layers) causes catastrophic forgetting while training only additive components preserves performance.
- The forward-switching mechanism ablation (Figures 6 and 7) provides convincing visual and quantitative evidence that separating output heads resolves the long-tail competition problem between 50,000 text tokens and 8,192 image tokens.
- The training cost represents a meaningful practical advantage.

**Key Weaknesses**

- The generation quality (GenEval 0.51) lags substantially behind state-of-the-art unified models such as Janus-Pro (0.80), D-DiT (0.65), and SynerGen-VL (0.61). The paper's characterization of this as highly competitive overstates the results.
- The interleaved generation evaluation (Table 4) compares against only two baselines (Anole and VARGPT). Additional interleaved-capable models such as Orthus, MiniGPT-5, and SEED-LLaMA are absent from comparisons. Additionally, multiple established interleaved generation benchmarks exist—ISG-Bench, InterleavedBench, OpenING, MMIE—but none are used.
- The paper claims interleaved generation but ARMOR currently generates at most one image per response (acknowledged only in Appendix B). This significantly limits the scope of the interleaved generation claim.
- The paper claims to offer a plug-and-play upgrade for various existing MLLMs but validates on only InternVL2.5-8B, with no experiments on other MLLM backbones.

**Audience:**

Yes

**Audience Explanation:**

The question of whether pre-trained MLLMs can be efficiently extended with new output modalities—without catastrophic forgetting—is of broad interest to the vision-language community. The upgrade paradigm addresses a practical need: many practitioners have access to strong understanding models but cannot afford to train unified models from scratch. The WoHG training strategy and parameter freezing ablations provide actionable insights for researchers working on continual learning, parameter-efficient fine-tuning, and multimodal model design. Even though the generation quality and interleaved evaluation have significant gaps, the core idea of extending existing expert models rather than rebuilding from scratch represents a direction that would interest researchers in this area.

**Broader Impact Concerns:**

The paper does not include a dedicated Broader Impact Statement in the main text. Given that ARMOR is a generative model capable of producing visual content, a Broader Impact Statement should be added. It should discuss the potential for generating misleading visual content and whether the base MLLM's safety alignment is preserved after the generative extension. The authors briefly mention NSFW filtering and domain bias in Appendix B, but this is limited to a few sentences within the Limitations section and does not constitute a sufficient treatment of ethical implications.

**Claims And Evidence:**

No

**Claims Explanation:**

The paper makes three central claims, and the evidence supporting them is uneven.

(1) **Catastrophic forgetting prevention**: This claim is the best supported. Table 2 shows ARMOR retaining most of InternVL2.5's understanding scores across nine benchmarks (approximately 95% on average), and the parameter freezing ablation in Table 5 systematically identifies which components cause degradation when unfrozen. The supplementary ablation comparing single-stage vs. multi-stage training (Tables 8-9) further validates the necessity of the WoHG approach. However, Table 5 only reports understanding metrics for Stage 1 configurations. Since different Stage 1 settings may propagate to affect final generation quality after Stages 2 and 3, the absence of end-to-end ablations (i.e., running the full 3-stage pipeline for each configuration and reporting both understanding and generation metrics) leaves the analysis incomplete. Additionally, the Switch column in Table 5 is insufficiently explained—experiments where Switch=× (Exp. 6, 7) show higher understanding scores, which could mislead readers without the clarification that these models cannot generate images at all and thus the higher score is a trivial result.

(2) **Competitive image generation at low cost**: The cost efficiency claim is well-supported by Table 3. The paper does acknowledge in Section 4.2 that a performance gap remains compared to top-tier specialized generation models. However, the abstract still uses the phrase highly competitive image generation, which is inconsistent with the actual results. More importantly, the gap is not only against generation specialists—ARMOR's GenEval of 0.51 also falls far below unified models such as Janus-Pro-7B (0.80), Janus-Pro-1B (0.73), D-DiT (0.65), and SynerGen-VL (0.61), which are the paper's most direct competitors. The abstract's language should be aligned with the more measured tone of the main text.

(3) **Accurate interleaved text-image generation**: This is the least convincingly supported claim, with several significant gaps.

- **Insufficient baselines**: Table 4 compares against only two models (Anole and VARGPT). Interleaved-capable models such as Orthus [1], MiniGPT-5 [2], SEED-LLaMA [3], and UnifiedGRPO [4] are not included.

- **No established benchmarks used**: Several well-established interleaved generation benchmarks exist—ISG-Bench [5], InterleavedBench [6], OpenING [7], MMIE [8], etc. The paper uses none of these and instead proposes its own metrics (Switch-Accuracy on 900 prompts and Interleave-CLIPScore on 300 prompts), which are not validated against human judgments.

- **Questionable metric design**: The ground-truth modality assignments for the 900 Switch-Accuracy prompts are not explained. The paper does not specify who determined that a given prompt requires an image vs. text response, nor what criteria were used for this assignment. Such modality labeling is inherently subjective, and without inter-annotator agreement or validation against human judgments, the reliability of this metric remains unclear.

- **Overstated scope**: ARMOR generates at most one image per response, yet the paper frames this as interleaved generation.


**Additional evidence concerns:**

- **Scaling claim**: Table 6 tests only two adapter sizes (2-layer and 4-layer) and claims ARMOR follows a favourable scaling law. Two data points are insufficient to establish a scaling law.

- **Missing comparison with MetaMorph**: The paper explicitly mentions MetaMorph in Appendix F as a model that also adopts the paradigm of extending pre-trained models with generation capabilities. Despite sharing the most similar motivation and approach with ARMOR, MetaMorph is absent from all experimental comparisons. This omission is notable given that MetaMorph represents the closest existing work to the proposed paradigm.

- **Generalizability**: The paper claims to provide a plug-and-play capability upgrade framework for various existing MLLMs but validates exclusively on InternVL2.5-8B. Moreover, part of the training data (ti2t subset) relies on distillations from InternVL-2.5 itself, meaning the data preparation pipeline is coupled to the specific base model. Switching to a different MLLM backbone would require re-generating this distillation data, which undermines the plug-and-play claim. Without experiments on at least one additional MLLM backbone, this generalizability claim is unsupported.


[1] Kou, Siqi, et al. "Orthus: Autoregressive interleaved image-text generation with modality-specific heads." arXiv preprint arXiv:2412.00127 (2024).

[2] Zheng, Kaizhi, Xuehai He, and Xin Eric Wang. "Minigpt-5: Interleaved vision-and-language generation via generative vokens." arXiv preprint arXiv:2310.02239 (2023).

[3] Ge, Yuying, et al. "Making llama see and draw with seed tokenizer." ICLR 2024.

[4] Nie, Ming, et al. "Towards Unified Multimodal Interleaved Generation via Group Relative Policy Optimization." NeurIPS 2025.

[5] Chen, Dongping, et al. "Interleaved scene graphs for interleaved text-and-image generation assessment." ICLR 2025.

[6] Liu, Minqian, et al. "Holistic evaluation for interleaved text-and-image generation." EMNLP 2024.

[7] Zhou, Pengfei, et al. "Opening: A comprehensive benchmark for judging open-ended interleaved image-text generation." CVPR 2025.

[8] Xia, Peng, et al. "Mmie: Massive multimodal interleaved comprehension benchmark for large vision-language models." arXiv preprint arXiv:2410.10139 (2024).

**Requested Changes:**

**Critical (required for acceptance)**

- Substantially strengthen interleaved generation evaluation. Evaluate on at least one established interleaved benchmark (ISG-Bench, InterleavedBench, OpenING, or MMIE). Add comparisons against interleaved-capable models such as Orthus, MiniGPT-5, SEED-LLaMA, and UnifiedGRPO. The current comparison against only Anole and VARGPT is insufficient.

- Adjust generation quality claims. Replace "highly competitive" in the abstract with language that accurately reflects the performance gap relative to Janus-Pro (0.80 vs. 0.51 GenEval) and other strong baselines. The abstract's language should be aligned with the more measured tone already present in Section 4.2.

- Clarify the scope of interleaved generation. The single-image-per-response limitation should be discussed in the main text (not only Appendix B). If the model can only produce text followed by one image, this is a meaningful limitation on the interleaved generation claim.

- Validate the Switch-Accuracy metric. Explain who assigned ground-truth modalities to the 900 prompts and by what criteria. Provide inter-annotator agreement or correlation with human judgments to establish the metric's reliability.

- Provide end-to-end ablation for Table 5. Currently, Table 5 only shows understanding metrics after Stage 1. Run the full 3-stage pipeline for at least the key configurations (Exp. 4 and 5) and report both understanding and generation metrics for the final model. Also explicitly explain the "Switch" column and why Switch=× experiments show higher understanding scores (because they lack generation ability entirely).

**Non-critical (would strengthen the paper)**

- Test on at least one additional MLLM backbone (e.g., Qwen series) to support the plug-and-play generalizability claim. Note that the current ti2t data pipeline relies on distillations from InternVL-2.5 itself, meaning the data preparation is coupled to the specific base model, further undermining the plug-and-play claim.

- Add a comparison with MetaMorph, which is explicitly mentioned in Appendix F as sharing the most similar paradigm of extending pre-trained models with generation capabilities.

- Support the scaling claim with more than two adapter sizes, or remove the favourable scaling law characterization from Section 4.3.

- Discuss why ARMOR's generation quality is substantially lower than Janus-Pro despite similar parameter counts. Is it primarily the 256×256 resolution, decoder capacity, training data volume, or a combination?

---

> ### Author Response · Authors · 2026-04-12
> **Rebuttal by Authors**
>
> ## Part1
>
> We sincerely thank the reviewer for the thorough and professional evaluation. These suggestions have been invaluable in improving our work. Below we address each point in detail. A revised PDF will be uploaded shortly, with all modifications highlighted in blue.
>
> ---
>
> ## Critical Changes
>
> ### Response to C1: Interleaved Generation Evaluation and Baseline Comparison
>
> We sincerely appreciate this constructive suggestion. We have conducted the following additional experiments:
>
> **(1)** We evaluate ARMOR on InterleavedBench, an established benchmark using GPT-4o for multi-dimensional assessment. The results are as follows:
>
> | Model | Text Quality | Perceptual Quality | Image Coherence | TIC | Helpfulness | AVG |
> |-------|:------:|:------:|:------:|:------:|:------:|:------:|
> | MiniGPT-5 | 1.22 | 2.45 | 1.62 | 2.03 | 1.77 | 1.82 |
> | GILL | 0.75 | 3.21 | 2.25 | 1.53 | 1.48 | 1.84 |
> | EMU-2 | 1.26 | 2.28 | 1.89 | 1.34 | 1.64 | 1.68 |
> | **ARMOR** | 3.67 | 4.01 | 1.93 | 4.17 | 2.92 | 3.34 |
> | Gemini1.5+SDXL | 4.40 | 3.99 | 3.64 | 4.13 | 3.62 | 3.96 |
> | GPT-4o+DALL·E 3 | 4.37 | 4.36 | 3.51 | 4.55 | 3.88 | 4.13 |
>
> ARMOR (AVG 3.34) outperforms all other evaluated end-to-end models, with Text Quality (3.67) and Text-Image Coherence (4.17) approaching those of pipeline-based systems. This is attributed to ARMOR inheriting the strong language understanding capability of its base MLLM.
>
> **(2)** We have added Orthus and MetaMorph as new baselines in Table 2 (understanding) and Table 3 (generation), as suggested by the reviewer:
>
> **Understanding (Table 2, unified models):**
>
> | Method | Par. | MMMU | MME-P | MMB | SEED | POPE |
> |--------|:----:|:----:|:-----:|:---:|:----:|:----:|
> | *Without interleaved output:* | | | | | | |
> | Show-o-256 | 1.3B | 25.1 | 948.4 | -- | -- | 73.8 |
> | SynerGen-VL | 2.4B | 34.2 | 1381.0 | 53.7 | 62.0 | 85.3 |
> | Janus-Pro | 7B | 41.6 | 1516.7 | 62.6 | 70.1 | 78.9 |
> | MetaMorph | 8B | 41.8 | -- | 75.2 | 71.8 | -- |
> | *With interleaved output:* | | | | | | |
> | Chameleon | 7B | 22.4 | 153.1 | 15.4 | 30.5 | 19.4 |
> | VARGPT | 9B | 36.4 | 1488.8 | 67.6 | 67.9 | 84.4 |
> | Orthus | 7B | 28.2 | 1265.8 | -- | -- | 79.6 |
> | **ARMOR** | 8B | 51.5 | 1635.2 |78.5 | 75.3 | 87.9 |
>
> **Generation (Table 3, unified models):**
>
> | Type | Method | #Param | #Train Images | Train Cost | GenEval↑ | FID↓ |
> |:----:|--------|:------:|:-------------:|:----------:|:--------:|:----:|
> | NoILO. | VILA-U | 7B | 15M | -- | 0.42 | 7.69 |
> | | Show-o | 1.3B | 36M | -- | 0.53 | 9.24 |
> | | D-DiT | 2B | 400M | -- | 0.65 | -- |
> | | TokenFlow-XL | 14B | 60M | -- | 0.55 | -- |
> | | SynerGen-VL | 2.4B | 667M | -- | 0.61 | 7.65 |
> | | Janus-Pro-7B | 7B | 72M | 3584/A100 | 0.80 | -- |
> | | Janus-Pro-1B | 1.5B | 72M | 1568/A100 | 0.73 | -- |
> | | SEED-X | 17B | 158M | ~960/A100 | 0.49 | 14.99 |
> | | MetaMorph | 8B | ~8M | -- | -- | 11.8 |
> | ILO. | Chameleon | 7B | 1.4B | 35687/A100 | 0.39 | -- |
> | | Orthus | 7B | -- | -- | 0.58 | -- |
> | | **ARMOR** | 8B | 5M | ~500/A100 | 0.51 | 9.07 |
>
> ARMOR achieves the best understanding performance among all models with interleaved output capability, while attaining competitive generation quality at a fraction of the training cost. It is worth noting that **both Orthus and MetaMorph, suggested by the reviewer, do not report quantitative results on any established interleaved generation benchmark**, whereas ARMOR additionally provides evaluation on InterleavedBench in the revised manuscript.
>
> ---
>
> ### Response to C2: Generation Quality Claims
>
> We appreciate this suggestion. We would like to clarify our positioning: **ARMOR's core contribution lies in demonstrating the feasibility of the "upgrade" paradigm, rather than pursuing state-of-the-art image fidelity.** The 256×256 resolution and lightweight decoder are deliberate design choices to efficiently validate this paradigm, and can be improved by adopting a higher-resolution VQ-VAE and scaling up training data. At roughly 1/70 of the training cost, ARMOR's GenEval of 0.51 and FID of 9.07 already demonstrate the effectiveness of this approach. We have revised the abstract wording to align with the more measured tone in Section 4.2.
>
>
> ---
>
> ### Response to C3: Single-Image Limitation in Main Text
>
> We appreciate this comment. We would like to clarify that **this is not an architectural limitation, but a scope decision.** As illustrated in Algorithm 1, ARMOR controls modality transitions via `<imgbos>` and `<imgend>` tokens, and the architecture natively supports multiple `<imgbos>`–`<imgend>` segments within a single sequence without any architectural modification. This work focuses on validating the core thesis—that a frozen MLLM can be upgraded into a unified model—and multi-image generation is a natural next step left for future work. We agree with the reviewer's suggestion and have explicitly discussed this point in the main text (Section 4.2) of the revised manuscript.

---

> ### Author Response · Authors · 2026-04-12
> **Rebuttal by Authors**
>
> ## Part2
>
> ---
>
> ### Response to C4: Reliability of the Switch-Accuracy Metric
>
> Thank you for this important question. The 900 prompts are sourced from **three inherently distinct datasets**: text-only prompts are sampled from the text-only conversation data in the LLaVA training set, image-only prompts from GenEval (a standard text-to-image benchmark), and mixed prompts from InterSyn. **The ground-truth modality of each prompt is determined by the inherent nature of its source dataset**—text-only conversations never involve image generation, GenEval prompts are exclusively image generation instructions, and InterSyn samples are natively in interleaved text-image format. This means the modality labels are deterministic and involve no subjective annotation ambiguity. We have added a description of the data sources in Section 4.1 of the revised manuscript.
>
> ---
>
> ### Response to C5: End-to-End Ablation for Table 5
>
> Thank you for raising this point. We believe there may be a misunderstanding. The purpose of Table 5 is to compare the impact of different parameter freezing strategies on understanding capability after Stage 1, in order to select the optimal configuration. **Exp. 5 is the configuration selected for the final ARMOR model, and all results in Tables 2, 3, and 4 are the end-to-end results of Exp. 5 after the full three-stage training pipeline.** Regarding the higher understanding scores of Exp. 6 and 7 (Switch=×), this is because they lack modality switching capability entirely and thus cannot perform any image generation—the higher understanding scores come at the cost of completely losing generation ability. To prevent similar misunderstandings, we have updated the caption of Table 5 in the revised manuscript with the above clarifications.
>
> ---
>
> ## Non-critical Changes
>
> ### Response to N1: Validation on Other MLLM Backbones
>
> All newly introduced components in ARMOR are attached via additive operations without modifying any parameters of the original model. This design is architecturally compatible with mainstream MLLMs that share the encoder-connector-LLM structure. Although ARMOR is a resource-efficient method, the full training pipeline still requires approximately 500 A100-days, making it infeasible to complete within the rebuttal period. **We are currently conducting validation experiments on InternVL3** and will include the results in a subsequent revision. We have adjusted the relevant claims in the revised manuscript to more accurately reflect the current scope of validation.
>
> Regarding the reviewer's concern about the ti2t data relying on distillation from InternVL-2.5, we conducted a verification experiment: we removed the approximately 25K distilled samples and re-ran Stage 1 training. The results are as follows:
>
> **Understanding capability:**
>
> | Method | Par. | MMMU | MME-P | MMB | SEED | POPE |
> |--------|:----:|:----:|:-----:|:---:|:----:|:----:|
> | ARMOR (w/o distill) | 8B | 50.7 | 1663.8 | 77.5 | 73.9 | 87.1 |
> | **ARMOR** | 8B | 51.5 | 1635.2 | 78.5 | 75.3 | 87.9 |
>
> **Modality switching capability:**
>
> | Model | Text Acc.(%) | Image Acc.(%) | Mixed Acc.(%) | CLIPScore |
> |-------|:------:|:------:|:------:|:------:|
> | ARMOR (w/o distill) | 94.1 | 90.2 | 90.9 | -- |
> | **ARMOR** | 94.9 | 88.7 | 91.5 | 35.6 |
>
> The results show that removing the distilled data causes only marginal changes in understanding capability, while modality switching remains unaffected. **The distilled data is not an essential component of the framework**; when adopting a different backbone, there is no need to rely on model-specific distillation, and this has no impact on the core contributions of this paper.
>
> ---
>
>
> ### Response to N2: Scaling Law Characterization
>
> We appreciate the reviewer's rigorous feedback on this point. We believe the original characterization of the experimental findings was imprecise, and have revised the wording from "favourable scaling law" to a more measured expression that better reflects the current scale of experiments.
>
> ---
>
> ### Response to N3: Analysis of Generation Quality Gap
>
> The gap primarily stems from three factors: (1) resolution (256×256 vs. 384 for Janus-Pro); (2) decoder capacity (lightweight VQ-VAE vs. larger decoders); and (3) training data scale (5M vs. 72M). **These are all engineering factors that can be improved through scaling**, and do not undermine the effectiveness of the upgrade paradigm proposed in this paper.
>
> ---
>
> ### Broader Impact
>
> We have added a Broader Impact paragraph in Appendix B, discussing the potential risks of generating misleading visual content and the design property that ARMOR preserves the original safety alignment of the frozen backbone.
>
> ---
>
> We sincerely thank the reviewer for the professional and constructive feedback, which has been instrumental in strengthening the paper. We will upload the revised PDF shortly, with all modifications highlighted in blue for easy reference.
>
> Best regards,
>
> The Authors

---

### Decision · Action_Editor_mgq5 · 2026-06-16

**Recommendation:** Accept with minor revision

**Additional Comments:**

Several concerns remain insufficiently addressed.

First, it is still unclear how the authors verify whether the existing understanding-specific image encoder preserves sufficient information required for image generation. If the encoder is originally optimized for understanding, it may discard or underrepresent details that are less important for VQA-style understanding but crucial for faithful image generation. Importantly, such information loss cannot be easily recovered by simply attaching a lightweight image decoder. Additional text-to-image experiments with longer and more compositional prompts, using relevant benchmarks such as GenEval++ and DPGBench, would help clarify whether the proposed framework can retain and utilize fine-grained generation-relevant information.

Second, the generality of the proposed framework across different backbone MLLMs remains insufficiently demonstrated. The authors mentioned that they would provide results with InternVL3, but it is unclear whether those results are included and how strong they are. Moreover, since InternVL3 still belongs to the InternVL family, experiments with a different backbone family would be important to support the claim that the proposed framework is broadly applicable beyond a specific model family.

Third, the necessity and effectiveness of the proposed three-stage curricular training algorithm, WoHG, require further clarification. A more direct baseline would be to train the model in a single stage using all training data and training tasks, while updating all components except the original text embedding layer and decoder layer. Especially, the current results in Tables 6 and 9 do not fully reveal the trade-off between preserving understanding capability and improving image generation or interleaved generation performance.

**Audience:**

Yes

**Audience Explanation:**

Building a unified MLLM is one of the most important and actively studied directions in recent AI. In particular, the proposed training framework presents a practically appealing approach for augmenting an existing understanding-oriented MLLM with generation capabilities at a low training cost, while largely preserving its original understanding capabilities. As such, the framework and the accompanying experimental results are likely to be of significant interest to researchers in this area as well as to the broader TMLR audience.

**Claims And Evidence:**

Yes

**Claims Explanation:**

This paper proposes a resource-efficient upgrade paradigm for building a unified multimodal LLM (MLLM) capable of multimodal understanding, text-to-image generation, and interleaved text-image generation, starting from a pre-existing understanding-oriented MLLM (InternVL2.5). Specifically, to preserve the original image understanding capabilities while extending the model toward generation, the proposed framework, ARMOR, introduces a lightweight image decoder on top of the original MLLM. It also employs a forward-switching mechanism that automatically selects the appropriate output modality through separate modality-specific heads. The model is trained through a three-stage progressive training strategy designed to gradually equip the original understanding model with image generation and interleaved generation capabilities while mitigating degradation in its original multimodal understanding performance.

Overall, the effectiveness of the proposed framework in acquiring new generation capabilities while preserving the original understanding capabilities, particularly with only a modest training cost, is well supported by extensive experiments and ablation studies on diverse benchmarks covering image understanding, including VQA, text-to-image generation, and interleaved text-image generation.

Nevertheless, there remain several limitations and concerns that have not been fully addressed, including in the authors’ responses. These issues are detailed in the additional comments below.

---

> ### Author Response · Authors · 2026-06-24
> **Response to the Action Editor**
>
> We thank the Action Editor for the constructive comments and the "Accept with minor revision" decision. We have revised the paper accordingly; all changes are highlighted in **blue** in the updated manuscript. Below we respond to the three points.
>
> ---
>
> ### **Comment 1** — Whether the frozen understanding-oriented encoder retains enough information for generation
>
> - Our experiments already provide direct evidence that the frozen, understanding-oriented backbone retains sufficient information to drive effective generation. ARMOR reaches a **GenEval score of 0.51 and an FID of 9.07**, producing good-quality images. In particular, although ARMOR reuses the image decoder of **Chameleon**, it clearly **surpasses Chameleon itself on GenEval (0.51 vs. 0.39)** — demonstrating that the information carried by the frozen backbone is enough for effective generation, not a bottleneck.
>
> - We have added a clarifying paragraph ("On the visual encoder and the scope of generation") in the Limitations section. We respectfully note that whether a stronger or more generation-specific encoder — or unfreezing and jointly training the encoder — could *further* improve generation is **orthogonal to the core scope** of this paper. We do not deny that such choices may bring additional gains, but they would substantially increase the training cost and thus run counter to our central motivation: **endowing a multimodal understanding model with generation capability in a lightweight, non-invasive manner while preserving its original understanding.** Maximizing absolute generation fidelity is not the goal of this work.
>
> ---
>
> ### **Comment 2** — Generality across backbone MLLMs (the promised InternVL3 results)
>
> - We have added the promised InternVL3 results to the paper (new **Table 7**, "Generality Across Backbones") together with the corresponding discussion. Applying the **identical** ARMOR framework and WoHG recipe to **InternVL3-8B** (a stronger, more recent backbone) preserves on average **~95% of the base model's understanding accuracy** across six benchmarks (MMMU 72.6→68.5, MME 2415→2337.4, MMVet 81.3→77.4, MMB 89.0→83.6, Hall 59.1→55.7, POPE 90.3→86.2), closely matching the retention observed on InternVL2.5-8B (~96%). This confirms that our core contribution does not hinge on a specific base model.
>
> - We acknowledge that InternVL3 still belongs to the InternVL family, and we have noted validation on other backbone families as future work.
>
> ---
>
> ### **Comment 3** — Necessity of the three-stage WoHG training
>
> - The single-stage baseline described — a single stage that updates all components **except** the original text embedding and decoder layers, on the full data/task mixture — is exactly the configuration already reported in the single-stage table in Appendix H (**Table 10** in the revised manuscript; this was Table 9 in the original submission, renumbered after we inserted the new backbone-generality table). We have clarified this in the revised version: the column previously labeled "Emb." is renamed **"Img. Emb."** (the newly added image-token embedding; the original text embedding and decoder layers are frozen throughout), and the caption/text now state explicitly that this is the natural single-stage baseline. This configuration **collapses understanding to as low as 48%** while the image branch fails to converge — i.e., it is an unfavourable trade-off in both directions and is strictly dominated by WoHG, which motivated our multi-stage design.
>
> - Regarding the trade-off in **Table 6**: this ablation deliberately isolates **Stage 1 ("what to generate")**, in which the image branch has **not yet been trained**, so image-generation metrics such as GenEval/FID are **not yet defined** there by construction. The table is intended to measure how training different modules interferes with **understanding**, while the "Switch" column captures the relevant Stage-1 capability. The full understanding–generation balance of the final three-stage model is reported in the main results (understanding benchmarks and GenEval/FID tables). We have added a sentence to the caption to make this explicit.

---

> > ### Comment · Action_Editor_mgq5 · 2026-07-16
> >
> > Some of my concerns have been addressed. However, the following issues still remain.
> >
> > - Even if the proposed framework primarily focuses on efficiently injecting image-generation capabilities, it is still necessary to evaluate whether the resulting model can generate images only from short and simple prompts or can also handle longer and more compositional instructions. Such an evaluation is essential for recent unified multimodal LLMs equipped with image-generation capabilities. Moreover, the paper explicitly states that “the fundamental hypothesis behind this decoupled architecture is that a powerful pre-trained encoder is sufficient to provide the rich semantic information required for generation.” Therefore, as requested in the original decision comments, I would still like the authors to report quantitative results on GenEval++ and DPGBench.
> >
> > - I appreciate that the authors conducted additional experiments using InternVL3-8B, even though it belongs to the same InternVL3 model family. However, since the image-generation performance of InternVL3-8B is not reported, the corresponding results should also be provided to demonstrate that the proposed framework can effectively inject image-generation capabilities into a different backbone.

---

> > > ### Author Response · Authors · 2026-07-19
> > >
> > > Dear Action Editor,
> > >
> > > Thank you for your further comments. While we were conducting the additional experiments in response to your comments, we received the notice that the deadline for the camera-ready submission had already passed. We have therefore submitted a camera-ready version at the same time. We hope that this does not violate the double-blind policy during the review period, and we kindly ask for your understanding.
> > >
> > > Following your guidance, we have additionally evaluated our model on GenEval++ and DPG-Bench, reported the results in the paper, and revised the corresponding descriptions accordingly; the related results and analysis are highlighted in blue. In addition, we have also recorded the image-generation performance of ARMOR (InternVL3-8B) in the paper.
> > >
> > > We hope these additions address your concerns, and we would be happy to make any further adjustments if needed.
> > >
> > > Best regards,
> > >
> > > The authors